# Photonic machine learning with on-chip diffractive optics

Tingzhao Fu[1], Yubin Zang[1], Yuyao Huang[1], Zhenmin Du[1], Honghao Huang[1], Chengyang Hu[1], Minghua Chen[1], Sigang Yang[1] & Hongwei Chen[1] ✉

Machine learning technologies have been extensively applied in high-performance information-processing fields. However, the computation rate of existing hardware is severely circumscribed by conventional Von Neumann architecture. Photonic approaches have demonstrated extraordinary potential for executing deep learning processes that involve complex calculations. In this work, an on-chip diffractive optical neural network (DONN) based on a silicon-on-insulator platform is proposed to perform machine learning tasks with high integration and low power consumption characteristics. To validate the proposed DONN, we fabricated 1-hidden-layer and 3-hidden-layer on-chip DONNs with footprints of 0.15 mm² and 0.3 mm² and experimentally verified their performance on the classification task of the Iris plants dataset, yielding accuracies of 86.7% and 90%, respectively. Furthermore, a 3-hidden-layer on-chip DONN is fabricated to classify the Modified National Institute of Standards and Technology handwritten digit images. The proposed passive on-chip DONN provides a potential solution for accelerating future artificial intelligence hardware with enhanced performance.

Concomitant with the substantial progress made in semiconductor technologies and novel computing architectures[1–9], artificial neural network (ANN)-related machine learning applications are being extensively utilized in many fields, including computer vision[10], natural language processing[11], emotion detection[12], speech recognition[13], medical image analysis[14,15], and decision-making[16,17]. However, to solve complex tasks in a timely manner, ANNs require massive amounts of resources, both regarding computing speed and energy consumption. In recent decades, optical neural networks (ONNs) have garnered tremendous interest, because of their advantages of low power consumption and ultrahigh computing bandwidth, which are unrivaled by their electronic counterparts[18–33]. Several implementations of ONNs have been proposed, including a coherent approach based on an integrated Mach–Zehnder interferometer (MZI) mesh[18,24,25,31], wavelength division multiplexing (WDM) processing with microring modulators, and programmable routing enabled by a phase-change material (PCM)[20]. However, these architectures are burdened by their limited computational scales, which are significantly restricted by their large footprint and energy consumption.

Recently, diffractive optical neural networks (DONNs) have garnered increased amounts of attention for their abilities to increase optical computing capacities and decrease power consumption levels by leveraging large-scale computations with the inherent parallel nature of optics[32,34–36]. This approach can map numerous neurons and connections onto optics, providing an even larger computational capacity than the conventional ONN architecture. However, mainstream DONNs are bulky because they are established on discrete diffractive components, causing significant difficulties integrating them into compact systems. In addition, complex calibrations between discrete devices may introduce additional errors.

In this work, we address the drawbacks of DONNs by proposing an on-chip DONN architecture based on an integrated one-dimensional (1D) dielectric metasurface. The 1D dielectric metasurface consists of a series of silicon slots filled with silicon dioxide; it represents the hidden

[1]Beijing National Research Center for Information Science and Technology, Department of Electronic Engineering, Tsinghua University, Beijing 100084, China. ✉e-mail: chenhw@tsinghua.edu.cn

layer (HL) in on-chip DONNs. To ensure that the pretrained parameters can be mapped accurately onto physical structures, a silicon slot group filled with silicon dioxide is used as a single neuron[22]. To demonstrate the capabilities of on-chip DONNs, we have fabricated an on-chip 1-hidden-layer DONN (DONN-I1) and an on-chip 3-hidden-layer DONN (DONN-I3) based on a silicon-on-insulator (SOI) platform to resolve the classification task on the Iris plants dataset[37]. The spacing between the adjacent HLs is set as 250 μm, and the footprints of the on-chip DONN-I1 and DONN-I3 are 0.15 mm² and 0.3 mm², respectively. The on-chip DONN-I1 and DONN-I3 yield accuracies of 86.7% and 90% for the blind test sets, respectively. Additionally, we propose an algorithm that is implemented through additional phase and power calibrations that compensates for the system errors caused by the chip fabrication and experimental implementation stages, which can increase the system noise resistance. In addition, to further verify the performance of the proposed on-chip DONN, we have designed a 3-hidden-layer DONN (DONN-M3) for the Modified National Institute of Standards and Technology (MNIST) classification task and obtained blind test set accuracies of 96.3% and 86.0% in numerical calculations and experimental tests, respectively. The aforementioned method for designing and fabricating on-chip DONNs, provides a solution for large-scale computation and overcomes the problem of complex alignment among the discrete components; these effects potentially pave the way for implementing future optical artificial intelligence accelerators, and they promote the potential application of photonic integrated devices in many other fields. The on-chip DONN architecture based on the standard complementary metal-oxide semiconductor (CMOS) process may realize low-cost mass manufacturing, providing a more realistic prospect for the large-scale commercialization of DONN chips in various applications.

## Results

### On-chip DONN model

The proposed on-chip DONN model consists of on-chip electromagnetic propagation, forward and error backward propagation, and a neuron-mapping process. The on-chip electromagnetic propagation model is modified based on the Huygens-Fresnel principle under restricted propagation conditions. It is an indispensable part of the on-chip DONN model, and can be described by Eq. (1):

$$w_{p,q}^m = \frac{1}{j\lambda} \cdot \left( \frac{1 + \cos\theta_{p,q}}{2r_{p,q}} \right) \cdot \exp\left( j\frac{2\pi r_{p,q} n_{\text{slab}}}{\lambda} \right) \cdot \eta \exp(j\Delta\phi) \quad (1)$$

where $m$ represents the $m$-th layer of the network, $p$ represents the $p$-th neuron at the position $(x_p, y_p)$ of layer $m$, $q$ represents the $q$-th neuron at the position $(x_q, y_q)$ of layer $m-1$, $\lambda$ is the working wavelength, $j = \sqrt{-1}$ represents an imaginary unit, $\cos\theta_{p,q} = (x_p - x_q)/r_{p,q}$, $r_{p,q} = \sqrt{(x_p - x_q)^2 + (y_p - y_q)^2}$ represents the distance between the $q$-th neuron in layer $m-1$ and the $p$-th neuron in layer $m$, $n_{\text{slab}}$ represents the effective refractive index (ERI) of the slab waveguide, $\eta$ represents a specific coefficient of the amplitude and $\Delta\phi$ represents a fixed phase delay[22]. The electric field evolution of the input signal propagation based on Eq. (1) is highly consistent with the simulation results of the 2.5D variational finite-difference time-domain (FDTD) solver (Supplementary Note 1.1).

By using an analytical expression of the on-chip electromagnetic propagation, the network structure parameters of the integrated DONNs can be pretrained via forward and error backward propagation algorithms (Supplementary Note 1.2). Once the parameters of the on-chip DONNs are determined, these parameters can be mapped onto physical structures, such as waveguides, grating couplers, multimode interferometer beam splitters, and silicon slot filled with silicon

dioxide (SSSD). Among the pretrained parameters, the physical neuron-mapping process is the most critical. To ensure the reliability of the mapping process, the pretrained phase value of a neuron is approximated by a slot group filled with silicon dioxide composed of more than two identical SSSDs. The length of the SSSDs in each group is calculated using Eq. (2):

$$L_{\text{slot}-i} = \frac{\Delta\varphi_i}{(n_{\text{eff}} - n_{\text{slab}}) \cdot k_0} \quad (2)$$

where $L_{\text{slot}-i}$ is the length of the SSSDs in the $i$-th group, $n_{\text{eff}}$ is the ERI of the slot group filled with silicon dioxide through which light passes, $n_{\text{slab}}$ is the ERI of the slab waveguide, $k_0 = 2\pi/\lambda$ is the wavenumber of light propagating in a vacuum, and $\Delta\varphi_i$ is the phase delay generated by the $i$-th slot group filled with silicon dioxide[22,38].

### DONN device architecture and design

In on-chip DONNs, as depicted in Fig. 1a, the trainable parameters are the phase values, which must be physically implemented by the diffractive units. Each diffractive unit (DU) is a slot group filled with silicon dioxide composed of three identical SSSDs; we record this slot group as a single neuron. For on-chip DONNs, the weight $W^{(k)}$ connecting each hidden layer is fixed, and trainable phase values on distinct HLs are achieved by designing the sizes of the DUs.

### Iris flower classifier

The on-chip DONN-I1 and DONN-I3 were designed and verified via a classification task on the Iris plants dataset. First, the input features were modulated onto the phase of the input light, and then the dataset coded in the optical phase was used to train the parameters of the on-chip DONNs by adopting the adaptive moment estimation (Adam) optimizer. Then, the pretrained parameters were mapped onto silicon-based structures (Supplementary Note 1.3). Additionally, to maximize the accuracy of the neuron-mapping process, the distances between the HLs were considered[22].

In this work, the proposed on-chip DONN was all-optical and used to solve complex tasks through the interference of transmitted light. The working wavelength of the laser was 1.55 μm. By fixing the width and thickness values of the SSSD to 200 nm and 220 nm, respectively, free control of the phase delays caused by the SSSDs were achieved within the range from 0 to 2π by changing the lengths of the SSSDs from 0 to 2.3 μm.

For the optimized on-chip DONNs in a classification task on the Iris plants dataset, the lengths of the HLs were 280 μm along the Y-axis; each HL contained 186 neurons and had 558 rectangular SSSDs. The distances between two successive HLs were 250 μm along the X-axis. The input signal was loaded onto the corresponding input waveguides and propagated 1010 μm through the inverse taper into the slab waveguide; then, the signal was propagated 250 μm through the slab waveguide to reach the first HL. After light exited the last HL, it propagated 250 μm until it reached the output layer of the network, with three detector regions (D₁, D₂, and D₃) arranged in a linear configuration. Each detector region was assigned a specific category. The width of each detector region was 8 μm, and the distances between the centers of two neighboring detector regions were 70 μm.

A schematic of the on-chip DONN-I3 is shown in Fig. 2. The on-chip DONN implemented inference and prediction mechanisms in a light-speed and passive manner; additionally, it could be applied in many fields, including computer vision, natural language processing, and image recognition. A conceptual diagram of the on-chip DONN application scenario is shown in Fig. 3.

### Numerical calculation and simulation

Based on the on-chip DONN model, the on-chip DONN-I1 and DONN-I3 were optimized and utilized for classification on the Iris plants dataset.

The dataset was divided into a training set and a testing set at a ratio of 8:2. The classification accuracies of on-chip DONN-I1 and DONN-I3 by numerical calculations were 86.7% and 90%, respectively. In addition, a 2.5D variational FDTD was used to verify the performance of the on-chip DONN-I1; the classification accuracy of the simulation result was 86.7%. The matching score of the classification predictions between the 2.5D variational FDTD and the numerical calculation was 100%. Figure 4 shows the simulation prediction process for the iris species and the corresponding output waveforms of the FDTD simulation. These theoretical studies included additional numerical calculation processes, relevant key parameters, and calculation results (Supplementary Note 2.1).

## Experiment

As a proof of concept, for the iris flower classifier, on-chip DONN-I1 and DONN-I3 were fabricated based on the SOI platform, and the micrographs are shown in Fig. 5a and Fig. 5c, respectively. After processing and testing, the chips were packaged to facilitate subsequent experiments (Supplementary Note 4.1). Experimental tests were performed on this basis (Supplementary Note 4.2). A laser with a working wavelength of 1.55 μm was coupled into the waveguide via an input grating coupler, and then the input signal was loaded onto the phase of the light through four phase shifters (PS). Finally, the modulated light interfered with the diffractive layers and was detected by the optical power meters at the output interface. The detected light was then transmitted to the central processing unit (CPU) through analog-to-digital conversion, as shown in Fig. 5d. Results of the numerical calculation and experimental implementation for on-chip DONN-I1 and DONN-I3 are listed in Table 1.

For the iris flower classifier, the testing accuracies of on-chip DONN-I1 and DONN-I3 without compensation were 56.7% and 60.0%, respectively, which were significantly different from the theoretical calculations of 86.7% and 90%, respectively. Phase errors are generated during the fabrication process, and error accumulation during light propagation significantly affects the performance of on-chip DONNs (Supplementary Note 3). Of course, in addition to the errors brought by the chip fabrication process, system errors could also be brought by the input signal loading and output signal detection stages during the experiment. Therefore, an algorithm compensation method consisting of phase compensation and power compensation was exploited to reduce the negative impacts of the errors (Supplementary Note 5.1 and Note 6). Moreover, the phase compensation stage was implemented based on the online in situ training procedure, during which a set of candidate voltage values can be obtained, and the output power was detected and recorded at this point. Here, the input signals were applied to the phases of light via the input voltages. After phase compensation, a traversal search method was adopted to find a set of optimal power compensation factors ($\alpha_1, \alpha_2, \alpha_3$) to maximize the prediction accuracy of the dataset. Consequently, when external algorithm compensation was employed, the experimental testing accuracy of on-chip DONN-I1 and DONN-I3 was improved to 86.7% and 90%, respectively. Figure 6 shows the experimental testing results of on-chip DONN-I1 and DONN-I3 before and after the introduction of the error compensation algorithm. From the compensated results, it can be observed that the compensation method is significantly effective, and the compensated results are consistent with the theoretical calculations.

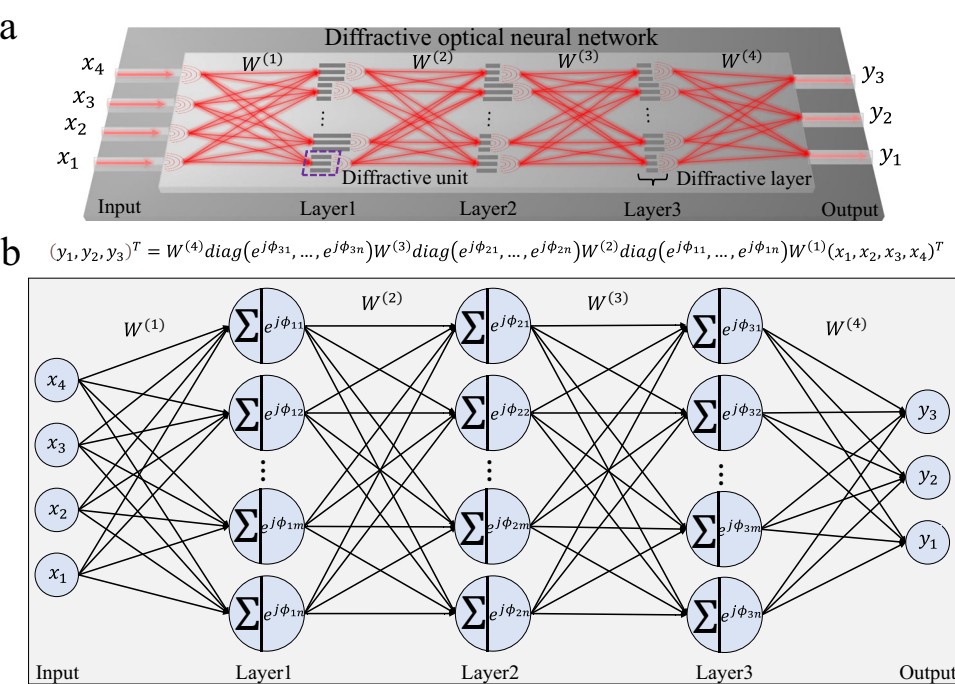

$$(y_1, y_2, y_3)^T = W^{(4)} diag(e^{j\phi_{31}}, \ldots, e^{j\phi_{3n}}) W^{(3)} diag(e^{j\phi_{21}}, \ldots, e^{j\phi_{2n}}) W^{(2)} diag(e^{j\phi_{11}}, \ldots, e^{j\phi_{1n}}) W^{(1)} (x_1, x_2, x_3, x_4)^T$$

**Fig. 1 | Schematic and logic diagram of on-chip diffractive optical neural network (DONN). a** Schematic of an on-chip DONN, each diffractive unit on a given layer acts as a secondary wave source, the amplitude and phase of which are determined by the product of the input wave and the complex-valued transmission at that unit. Each diffractive unit (DU) is a slot group composed of three identical silicon slots that are filled with silicon dioxide; each DU represents a single neuron in the on-chip DONN. **b** Logic diagram of Fig.1a that mathematically describes the physical calculation process of the on-chip DONN. The formula shown between Fig. 1a and Fig. 1b is the mathematical expression of DONN, where "T" represents matrix transposition; $diag(e^{j\phi_{11}}, \cdots, e^{j\phi_{1n}})$, $diag(e^{j\phi_{21}}, \cdots, e^{j\phi_{2n}})$, and $diag(e^{j\phi_{31}}, \cdots, e^{j\phi_{3n}})$ refer to a diagonal matrix, that is, a matrix in which the elements outside the main diagonal are all 0, where the phase values $(\phi_{11}, \cdots, \phi_{1n}, \phi_{21}, \cdots, \phi_{2n}, \phi_{31}, \cdots, \phi_{3n})$ are generated by the corresponding DUs; $W^{(k)}$ represents the $k - th$ diffraction matrix derived from the on-chip electromagnetic propagation model (Eq. (1)); $(x_1, x_2, x_3, x_4)$ represents the input; and $(y_1, y_2, y_3)$ represents the output.

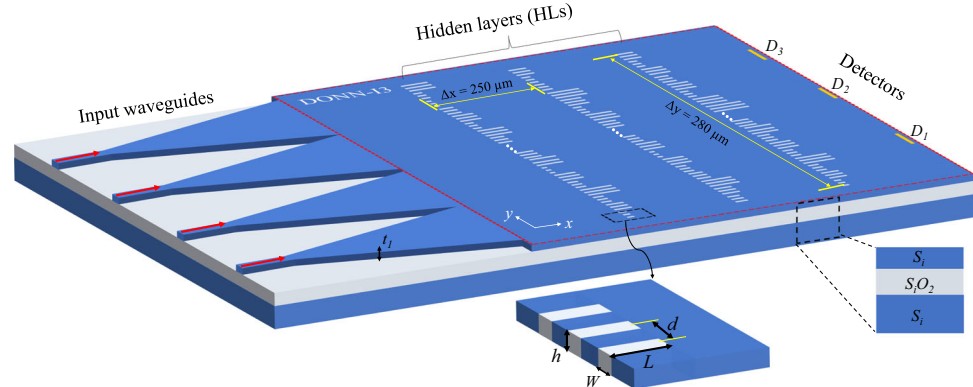

**Fig. 2 | Schematic of the on-chip DONN-I3 structure.** The schematic includes three hidden layers, and each neuron in the hidden layer consists of three identical silicon slots filled with silicon dioxide representing a complex-valued transmission coefficient. The transmission coefficients of each layer are trained by using deep learning to perform a function between the input and output planes of the network. Then, following the fabrication of the on-chip DONN, the DONN performs the learned function at the speed of light in a passive manner. The center distances between the adjacent slots $d$ are 500 nm, the periods of the silicon slot groups filled with silicon dioxide are 1.5 μm, the width of the slot $w$ is 200 nm, the thickness of the slot $h$ is 220 nm, and the length of the slot $L$ is determined by the pretrained phase values based on Eq. (1), and the thickness of the light propagation layer $t_1$ is 220 nm. The distances between adjacent HLs are 250 μm along the X-axis, and the lengths of each HL are 280 μm along the Y-axis.

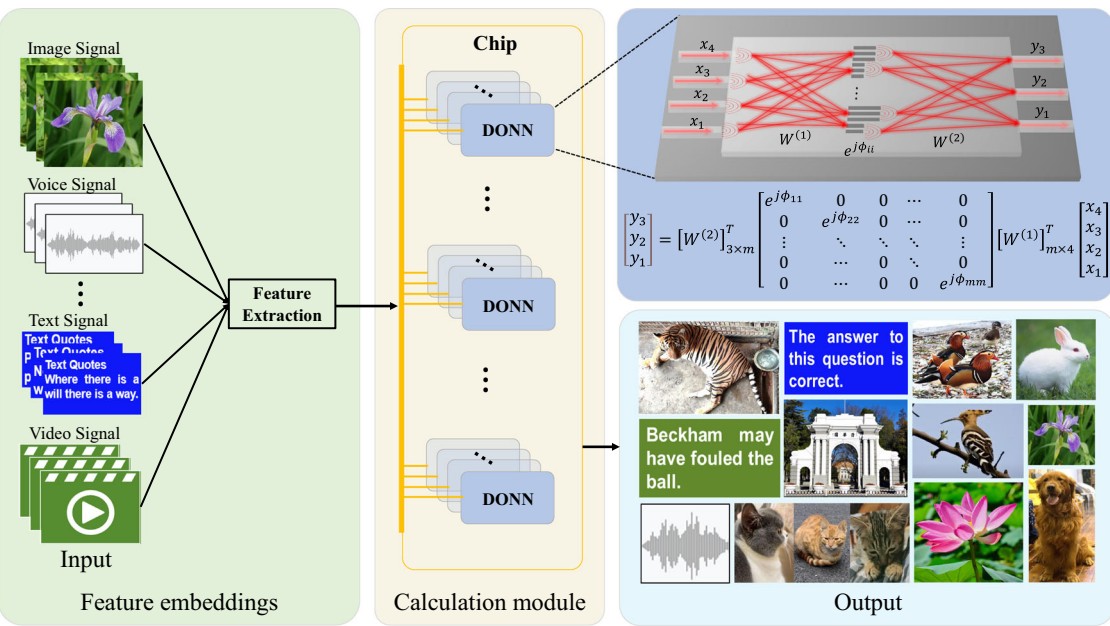

**Fig. 3 | Conceptual diagram of multichannel on-chip DONNs for various tasks.** The features of different signals are extracted and encoded onto the phase, amplitude, or polarization of light. Then, the input signals containing optical information are fed into the on-chip DONNs for subsequent calculations.

## Further experimental verification

Based on the same design principle of the iris flower classifier, a more complicated dataset—the Modified National Institute of Standards and Technology (MNIST) handwritten digit images—is used to validate the functionalities of our proposed on-chip DONNs. The MNIST dataset is split into training (60,000 images) and testing sets (10,000 images). In this work, for the handwritten digit classifier, the input $28 \times 28$ grayscale image is reshaped into a $784 \times 1$ vector and compressed into 10 features through a full connection layer network.

For the optimized on-chip DONN-M3, the lengths of the HLs were 105 μm along the Y-axis; each HL contained 70 neurons (consisting of 210 rectangular SSSDs). The distances between two successive HLs were 250 μm along the X-axis. The ten input features were loaded onto the ten corresponding input single-mode waveguides and propagated directly into the slab waveguide, and then propagated 250 μm through the slab waveguide to reach the first HL. After light exited the last HL, it propagated 250 μm until it reached the output layer of the network; the output layer featured ten detector regions D$i$ ($i=1,2,\ldots,10$) arranged in a linear configuration. Each detector region was assigned a specific category. The width of each detector region was 8 μm, and the distances between the centers of the two neighboring detector regions were 8 μm.

The numerical calculation accuracy of on-chip DONN-M3 for the 10000 blind testing sets was 96.3%. We randomly selected 100 handwritten digits from the 10,000 blind testing sets for experimental verification, achieving a classification accuracy of 86.0% under the external error compensation scenario. The relevant pictures during the packaging process of the on-chip DONN-M3 are shown in Fig. 7a–c. The micrograph of the on-chip

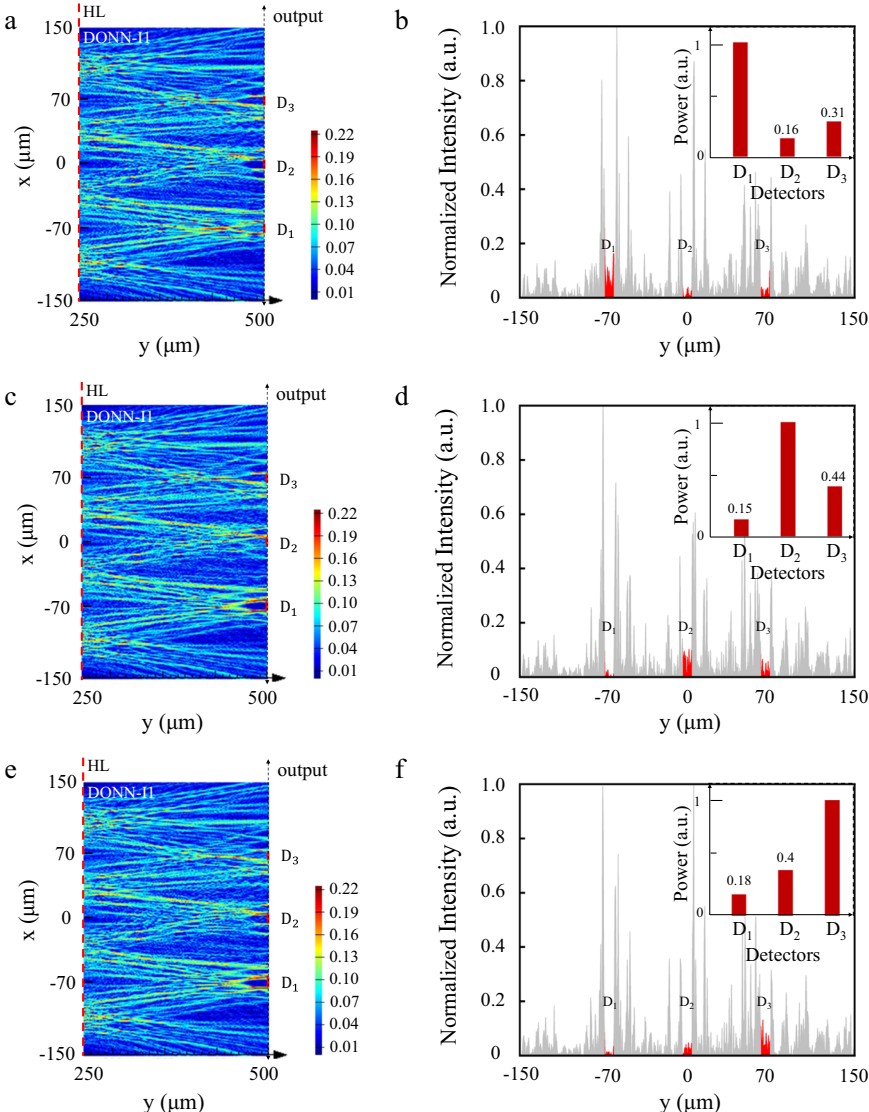

**Fig. 4 | Simulation results of the proposed on-chip DONN-I1.** $D_1$, $D_2$, and $D_3$ represent the prediction of the different kinds of iris flowers, i.e., Setosa, Versicolor, and Virginica. The red dotted line in the figure indicates the location of the hidden layer (HL). **a** Prediction of Setosa; in other words, the power of $D_1$ should be greater than that of $D_2$ and $D_3$. Here, the prediction result is correct. **b** Output waveform of (**a**), where the red marks are the detection areas, with each detector 8 μm wide and the center spacing between adjacent detectors is 70 μm. **c** Prediction of Versicolor; the power of $D_2$ is greater than that of $D_1$ and $D_3$. Thus, the prediction result is correct. **d** Output waveform of (**c**). **e** Prediction of Virginica; the power of $D_3$ is greater than that of $D_1$ and $D_2$; thus, the prediction result is correct. **f** Output waveform of (**e**); where, the unit of the normalized power 'a.u.' is the abbreviation of 'arbitrary unit'.

DONN-M3 structure is shown in Fig. 7d, and the close-up taken by scanning electron microscopy (SEM) of the diffractive units is shown in Fig. 7e. The confusion matrix of the experimental testing result is shown in Fig. 7f. The recognition results of hand-written digits 3, 5 and 9 after system error compensation are shown in Fig. 7g–i, respectively (more details are described in Supplementary Note 2.2 and Note 4.3).

## Discussion

According to the experimental results, Table 1 indicates that the prediction accuracies of the on-chip DONN-I1 and DONN-I3 without algorithm compensation on the Iris plants dataset are 56.7% and 60.0%, respectively; these values are quite different from the numerical calculation results of 86.7% and 90%, respectively. By assuming that the differences between the experimental and numerical calculation results are attributable to the errors caused by the fabrication process, the working processes of the on-chip DONNs are analytically

expressed by Eq. (3) and Eq. (4):

$$Y_{cal} = D_{cal}X \tag{3}$$

$$Y_{chip} = D_{chip}X \tag{4}$$

where $Y_{cal}$ is the theoretical calculation result of the product of input $X$ and the transfer matrix $D_{cal}$, $D_{chip}$ is the transfer matrix of light propagating in the slab waveguide, and $Y_{chip}$ is the output electric field of the product of input $X$ and the transfer matrix $D_{chip}$. Due to inevitable machining errors, the error transfer matrix $D_{err}$ will exist naturally after fabrication, and mathematically, $D_{err}$ is the difference between $D_{cal}$ and $D_{chip}$. Furthermore, $D_{err}$ results in the difference $P_{err}$ between the theoretically calculated power $P_{cal}$ and the detected power $P_{chip}$; that is, $P_{err} = P_{cal} - P_{chip}$. External algorithm compensation aims to find a set of input voltage values and power compensation factors $\alpha$ that

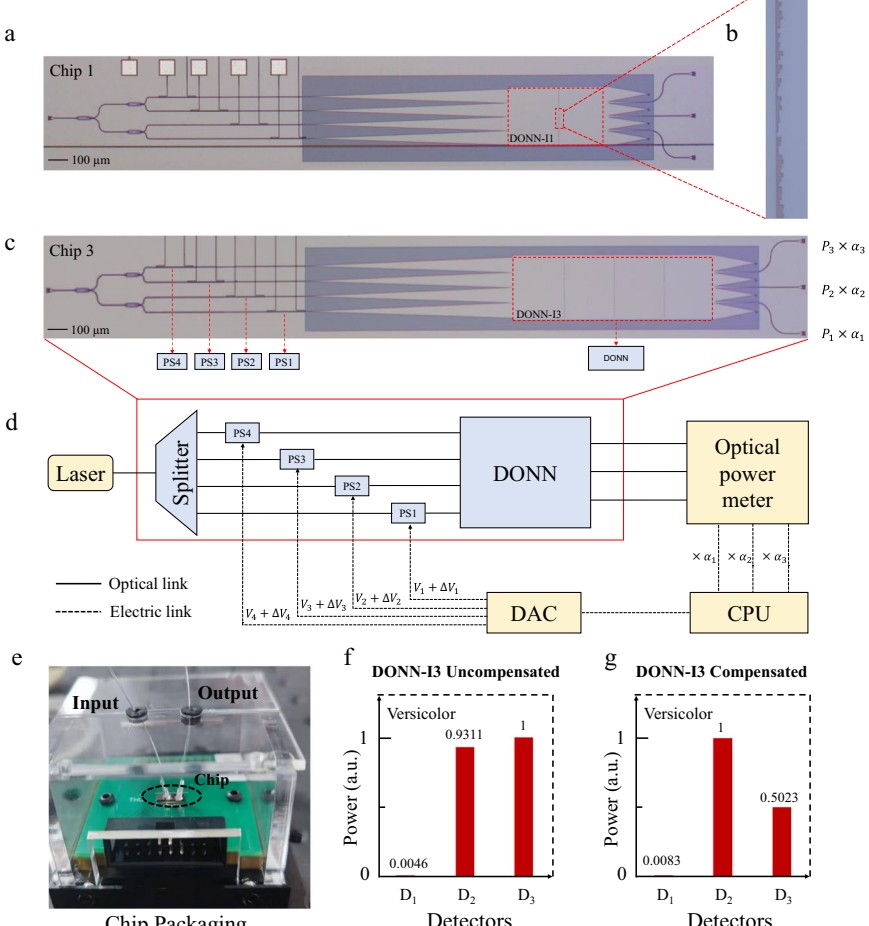

**Fig. 5 | Structures of the on-chip DONN-I1 and DONN-I3 specimens and the experimental flow and test results. a**, **c** Micrographs of on-chip DONN-I1 (**a**) and DONN-I3 (**c**). **b** Close-up view of the partial slot array under the microscope's 100 × lens. **d** Experimental compensation testing process, where $V_i + \Delta V_i$ is the input voltage used to load the input signal and compensate for system errors. $V_i$ is the loading voltage corresponding to the original signal of the dataset, $\Delta V_i$ is the compensation voltage found through the optimization algorithm in the phase compensation process, and $\alpha_i$ is a compensation factor for the output power to compensate for the system errors. **e** Diagram of the chip after packaging.

**f**, **g** Experimental testing results for a sample in the testing sets before (**f**) and after (**g**) compensation. The input features of the sample belong to Versicolor; thus, its correct prediction result is that the power of $D_2$ should be the largest. **f** and **g** show that after compensation, the prediction result is corrected, from the wrong result being (**f**) to the correct result being (**g**); where, the unit of the normalized power 'a.u.' is the abbreviation of 'arbitrary unit'; CPU is the abbreviation of central processing unit; DAC is the abbreviation of digital to analog convertor; PS is the abbreviation of phase shifter; $P_1$, $P_2$ and $P_3$ represent the optical power detected by different output ports respectively.

minimize the difference $P_{\text{err}}$; that is, the algorithm seeks to minimize the absolute value $|P_{\text{cal}} - \alpha \odot P_{\text{chip}}|$ (where $\odot$ indicates multiplication with the corresponding element); for example, the optimal solution $|P_{\text{err}}| = |P_{\text{cal}} - \alpha \odot P_{\text{chip}}| \approx (0, \ldots, 0)^T$. In this experiment, after compensation by the algorithm, the prediction accuracies of the on-chip DONN-I1 and DONN-I3 were improved to 86.7% and 90%, respectively, which are well consistent with the theoretical calculation results. In addition to the error brought by the chip fabrication stage, the additional errors in the system would also be caused in the input signal loading and output signal detection stages. Therefore, an effective external compensation algorithm is significant for the overall system

error correction and compensation. (Supplementary Note 5.1 and Note 6). It is worth noting that when the system error is more complicated, the higher error correction capability of the error compensation algorithm is needed; for example, in the further experimental verification (for the handwritten digit classifier), a $10 \times 10$ full connection layer after the DONN-M3 chip is trained to realize the system error compensation (Supplementary Note 5.2).

The error in the system mainly comes from three aspects: the signal loading, chip fabrication, and signal detection stages. In future work, several methods can be used to reduce system errors. First, through using more advanced machining equipment, which can fundamentally reduce the error caused by chip processing. Second, the random phase offset with uniform distribution within the interval, such as $(0, 0.5\pi)$, can be introduced to each part during the training stage, such as the signal loading and fabrication stages, to improve the system's robustness against nanofabrication variations and phase fluctuations in measurement[33]. Last but not least, it is extraordinarily significant to further improve the resolution of the testing instrument and the stability of the testing environment to ensure that the error caused by the testing process is minimized.

**Table 1 | Numerical calculation and experimental testing results for on-chip DONNs**

| Accuracy\--- | Numerical calculation | Experimental test | |
|---|---|---|---|
| | | Uncompensated | Compensated |
| On-chip DONN-I1 | 86.7% | 56.7% | 86.7% |
| On-chip DONN-I3 | 90.0% | 60.0% | 90.0% |

To illustrate the effectiveness and importance of the pretrained parameters of HL, 10 groups of HL parameters for different on-chip DONN-I1 and DONN-I3 are randomly generated, and the Iris plants dataset is used to test the performance levels of these on-chip DONNs. The prediction accuracy results are shown in Fig. 8a and Fig. 8b. The prediction results of the on-chip DONNs with pretrained HL parameters (serial number 1) are significantly higher than those of the on-chip DONNs with randomly generated HL parameters (serial numbers 2–11). The results prove that the pretrained parameters are imperatively significant and effective.

For computation speed and energy efficiency, when handling complex tasks, such as automatic driving and real-time missile tracking, ANNs with high speed and low energy consumption are necessary. Our on-chip DONN architecture takes advantage of processing big data at high speeds and low power consumption. Once all the parameters have been trained and mapped onto physical structures, forwards propagation computing is performed optically on a passive system. Assuming that our on-chip DONN has $N$ neurons at each HL, implementing $m$ layers of $N \times N$ matrix multiplication and operating at a typical 100 GHz photodetection rate[39,40], the number of floating-point operations per second (FLOPS) to match the optical network is obtained using Eq. (5):[18]

$$R = 2m \times N^2 \times 10^{11} \, \text{FLOPS} \qquad (5)$$

where $R$ is the number of operations per second (the time it takes from receiving input signals to computing an inference result, without considering the time spent in the signal loading stage), this value is related to the number of $N \times N$ matrix, the number of neurons on each HL, and detection rate of the photodetectors. Therefore, for the on-chip DONN-I3, the computation speed is approximately $1.38 \times 10^{16}$ FLOPS, as calculated by Eq. (5), this value is four orders of magnitude higher than the performance levels of modern graphic processing units (GPUs), which typically perform at $10^{12}$ FLOPS[25]. Moreover, in the optical calculation process, the calculation delay was approximately 27.56 ps (Supplementary Note 7.1). Regarding energy consumption, the input power of the laser under 1.55 μm is 32 mW. The input signal is loaded by the thermo-optic phase shifters, and the average energy required to set each phase shifter to 2π rad is approximately 30 mW. The calculation process of the computing part is fully passive, thus the energy consumed to complete one calculation for the proposed on-chip DONN-I3 system is approximately $1.1 \times 10^{-17}$ $J$/FLOP. (Supplementary Note 7.2).

To date, the scalability of on-chip neural networks is an obstacle. For example, interference ONNs based on MZIs[18] and pulse ONNs based on microring resonators (MRRs)[20] cannot dramatically expand the number of neurons due to the large footprint of each device. The on-chip DONN is a feasible method for solving this problem. In this work, we design an on-chip DONN-I3 with three HLs; each HL includes 186 neurons. Through the recent design method, approximately 2000 neurons can be designed per square millimeter. Once the neuron mapping method further improves, the number of integrated neurons can significantly increase. For the reconfigurability of on-chip DONNs, PCM materials are candidates for future studies. For example, related works on PCM material for realizing reconfigurable networks have been reported[20,41].

For the performance of the proposed DONN framework, Table 2 shows a comparison of the designed on-chip DONN-I3 with other integrated ONNs. The matrix dimension is a key parameter for handling complex tasks; the size of the matrix dimension depends on the number of integrated neurons. For more complicated tasks, the energy demand in the calculation process is greater. Therefore, a passive calculation process is imperative. From a comprehensive perspective, our proposed on-chip DONN architecture is a notable choice.

To conclude, fully optical on-chip DONNs based on the SOI platform are proposed and fabricated in this work. On-chip DONNs can perform complicated functions at faster speed and with lower latency and power consumption levels than conventional ANNs. Inference tasks are used to demonstrate the performance levels of the on-chip DONNs; the results are excellent after introducing a compensation algorithm. Note that, nonlinear activations are only used in the output layer in this study, and the results for inference tasks will improve if nonlinear activation functions are considered in each hidden layer of the on-chip

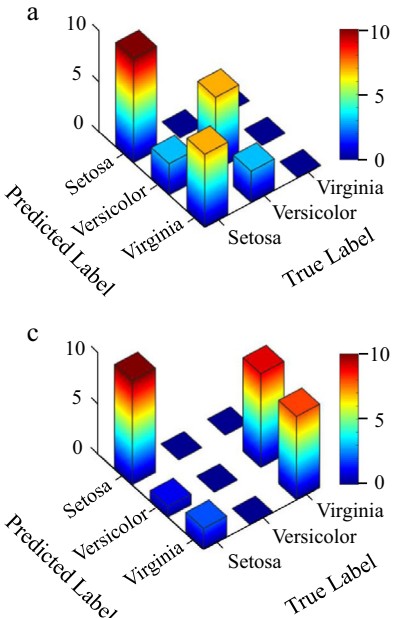

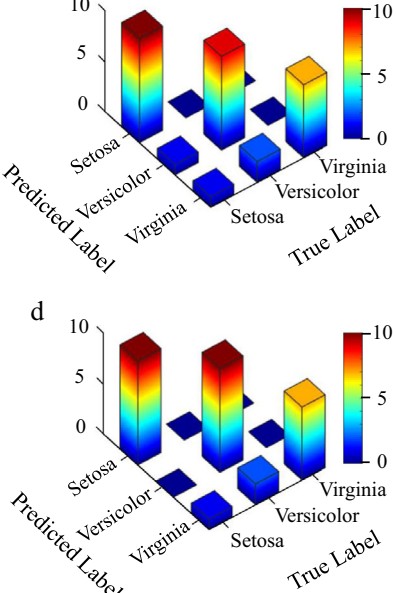

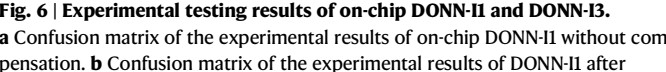

**Fig. 6 | Experimental testing results of on-chip DONN-I1 and DONN-I3.** **a** Confusion matrix of the experimental results of on-chip DONN-I1 without compensation. **b** Confusion matrix of the experimental results of DONN-I1 after compensation. **c, d** Confusion matrixes of the experimental results of on-chip DONN-I3 before (**c**) and after (**d**) compensation.

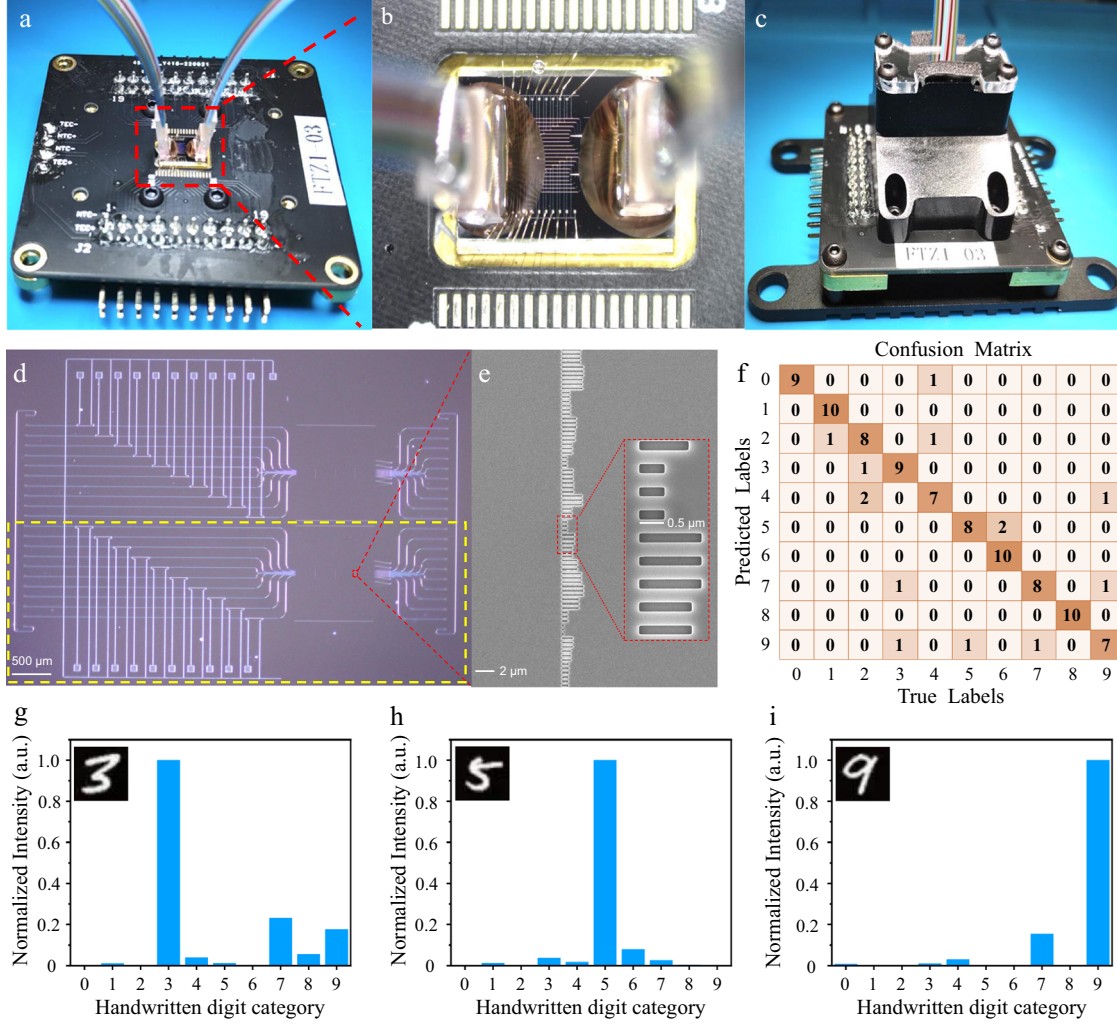

**Fig. 7 | Structure of on-chip DONN-M3 and the experimental flow and test results. a, b**, and **c** Pictures after wiring and packaging. **d** Micrograph of the on-chip DONN-M3, in which the yellow dotted line frame part is the chip structure of this experiment. **e** SEM image of the diffractive units. **f** Confusion matrix of the experimental results of on-chip DONN-M3. **g, h**, and **i** Recognition results of handwritten digits 3, 5 and 9, respectively; where, the unit of the normalized power 'a.u.' is the abbreviation of 'arbitrary unit'.

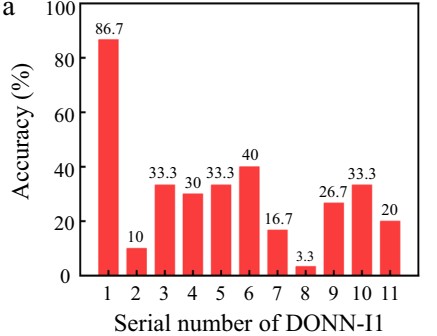

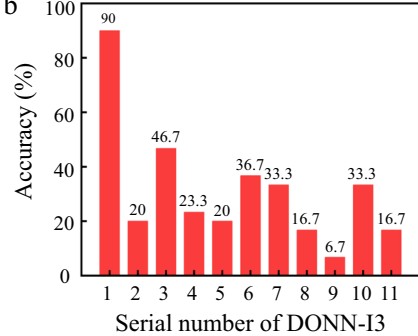

**Fig. 8 | Effectiveness validation of the pretrained parameters.** Prediction accuracy results for **a** the on-chip DONN-I1 and **b** the on-chip DONN-I3. Serial number 1 indicates the result obtained on the premise of pretrained hidden layer (HL) parameters. Serial numbers 2–11 indicate the results obtained on the premise of randomly generating HL parameters.

DONN system. Consequently, we will consider the implementation of nonlinear functions on-chip in combination with PCM in future works. Furthermore, relative to other ONNs, the proposed on-chip DONN has the advantages of a simple structure design, all-optical passive operation, and massive-scale neuron integration. This on-chip DONN architecture is a potential solution for accelerating future artificial intelligence hardware with enhanced performance levels.

**Table 2 | Comparison of the on-chip DONN-I3 and other integrated ONNs**

| Works\ Index | Basic units | Integration in theory (NBUs/mm²) | Operational power (J/FLOP) | Throughput (TOPS) | Verification method |
|---|---|---|---|---|---|
| Ref. 18 | MZI | <10 | $7.66 \times 10^{-14}$ | 6.4 | Experiment |
| Ref. 42 | MZI | <10 | $2.14 \times 10^{-13}$ | 21.6 | Experiment |
| Ref. 43 | MZI + diffractive cell | <20 | $1.41 \times 10^{-15}$ | 32 | Experiment |
| Ref. 44 | MMI | <25 | $3.07 \times 10^{-14}$ | 30 | Experiment |
| Ref. 23 | MRRs covered with PCM | <5 | $5.9 \times 10^{-15}$ | 28.8 | Experiment |
| Ref. 22 | SWU | $\sim9.5 \times 10^{3}$ | N/A | $1.96 \times 10^{3}$ | Simulation |
| Ref. 21 | SWU | $\sim3.75 \times 10^{4}$ | N/A | $1.54 \times 10^{4}$ | Simulation |
| Our work | SWU | $\sim2 \times 10^{3}$ | $1.1 \times 10^{-17}$ | $1.38 \times 10^{4}$ | Experiment |

In Table 2, MZI Mach–Zehnder interferometer, MRRs Microring resonators, MMI Multimode interferometer, SWU Subwavelength unit, NBUs Number for basic units, FLOP floating-point operation, and TOPS Trillions ($10^{12}$) of operations per second. The operational power only refers to the energy consumption of the computing units, the laser, and the signal loading stage when the system is working, excluding the energy consumption of the signal detection and peripheral driving stages. The throughput is uniformly calculated according to Eq. (5).

## Methods

### Device fabrication
The entire on-chip DONN was fabricated on an SOI (100 substrate) platform with a 220 nm thick silicon (Si) top layer and a 3 μm thick buried oxide. For the on-chip DONN-I1 and DONN-I3, slots were created by etching the 220 nm Si film layer; then, a 2 μm thick silicon dioxide (SiO$_2$) upper cladding was deposited on the Si film layer. Next, a thin layer of titanium nitride (TiN) was deposited as a resistive layer for the heaters, and a metal film of AlCu (Cu:0.5%) was patterned as the electrical connection to the electrodes and heaters. Finally, a 2 μm thick silicon dioxide (SiO$_2$) protection layer was deposited on the device layer. For the on-chip DONN-M3, slots were created by etching the 220 nm Si film layer, and then a 2 μm thick silicon dioxide (SiO$_2$) upper cladding was deposited on the Si film layer. Furthermore, a thin layer of titanium (Ti) was deposited as a resistive layer for the heaters, and a metal film of aluminium (Al) was patterned as the electrical connection to the electrodes and heaters. Finally, an 800 nm thick silicon dioxide (SiO$_2$) protection layer was deposited on the device layer.

### Optical measurements
A continuous-wave tunable semiconductor laser with a polarization controller was used to launch light onto the chip (15 dBm). The fiber-grating coupler loss was optimized to 5 dB per input/output facet for the on-chip DONN-I1 and DONN-I3 chips, and 6.5 dB per input/output facet for the on-chip DONN-M3 chip. The outputs were monitored using the multichannel optical power meters; the minimum power detection limit was −75 dB. An external auxiliary circuit was provided by a DC dual-tracking voltage-stabilizing source (DH1718E-5, 0–35 V).

### Numerical simulations
The training process of the iris flower classifier and the handwritten digit classifier were conducted in PyTorch, which is a package for Python. The light diffraction connection in the process of forward and error backward propagation followed the modified Huygens-Fresnel principle. The input features were encoded into the light phase, ranging from 0 to $2\pi$. A 2.5-dimensional variational FDTD method (http://www.lumerical.com/tcad-products/fdtd/) was used to simulate the optical field distribution and the on-chip DONN-I1 system. A conformal mesh with a spatial resolution of less than 1/10 of the smallest feature size was applied.

## Data availability
The data that support the findings of this study are available from the corresponding authors on request.

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

## Acknowledgements

This work was supported by the National Key Research and Development Program of China (2019YFB1803500) and the National Natural Science Foundation of China (NSFC) (62135009).

## Author contributions

T.F. and H.C. conceived the idea. T.F. developed the design principle and numerical simulations. T.F., Y.Z. and Z.D. performed the experiments. T.F., Y.Z., H.H., Y.H., Z.D., C.H., S.Y., M.C. and H.C. involved in the discussion, theoretical analysis and data analysis. T.F. prepared the manuscript. Y.H., Y.Z., and H.C. revised the manuscript. H.C. supervised and coordinated all the work. All authors commented on the manuscript.

## Competing interests

The authors declare no competing interests.
