## [Peer Review File · Nature Communications]

Photonic machine learning with on-chip diffractive opticsREVIEWER COMMENTS

Reviewer #1 (Remarks to the Author):

The authors proposed a on-chip diffractive optical neural network (DONN) based on a silicon-on-insulator (SOI) platform to achieve machine-learning computing with light-speed, high integration, and efficiency. The authors fabricated 1-16 hidden-layer and 3-hidden-layer on-chip DONNs and perform a classification task on the Iris dataset with the accuracies of 86.7% and 90%. Although the authors have obtained good classifying performance, I strongly concern the novelty and significance of the manuscript. In fact, the proposed on-chip DONN architecture based on the integrated one-dimensional dielectric metasurface is very similar to that in [Ref. 33]. The novelty declared by the authors are mainly compared to D2NNs based on 3D printing and diffractive propagations in space. However, there have been a lot of researches on planar optical D2NN and on-chip D2NN. The authors do not address these highly related works to illustrate the advantages of this work. Hence I suggest to reject its publication in Nature Communications.

Besides, the following concerns should be addressed.

1. The simulation results presented in Figure 4(b)(d)(f) show excessive peaks at the output plane, especially in the proximity of the detectors, which may hinder the system performance considering fabrication margin. I would like to know would the unwanted peaks interference with the detector outputs and could they be eliminated in the training process of the proposed DONN?
2. The Iris plants dataset is a widely used benchmark for simple machine learning algorithms, which is easily to achieve high classification accuracy because the feature of input values could be clustered at low feature plane. MNIST handwritten digit database is suggested to be numerically demonstrated.
3. The 1D nature of the proposed platform highly limits the classification of high-resolution images. Can the authors discuss this limitation and compare it with 2D metasystems?
4. I believe that the real system based on the proposed DONN could not realize the theoretical speed given in Line 261. Though the photodetection rate and the equivalent processing speed on the platform of DONN exhibit unlimited potentials, I am afraid that the bottleneck is the digital-to-analog conversion (DAC) stage that modulate the signal for input vector injection. Could the author present some discussion relating to this problem?
5. Formulas and figures should be revised for a better comprehension. For example, in Equation (1), the transmission parameter w_i^m would better be replaced by w_{ij}^m ; in Figure 1, the matrix and vectors should be marked accordingly for a contrast with the scalar variables; in Figure (6), 'Truth Label' and 'Predicted Label' should be added at the corresponding axis.

Reviewer #2 (Remarks to the Author):

I am in principle supportive of the neat concept the authors have presented in this paper. However, I have serious concerns on the objectivity of the results as a consequence of the error compensation algorithm. Before recommending the publication of this paper, such comments need to be addressed, emphasizing again on demonstrating that the error compensation algorithm is not arbitrarily manipulating the result of the computation. Please see my detailed comments below.

On error compensation

- I am concerned about the way the error compensation algorithm works. For once, the fact that the algorithm allows the authors to reach, exactly, the same values calculated in simulations is somehow a red flag. I can see how the phase errors introduced by fabrication defects can be compensated, but my main problem is the power compensation on the output signals. Since the inference is performed using intensity measurements, a direct manipulation of such intensity is a manipulation of the

computing output. Multiplying the three output channels by optimized weights seems like a way to get any expected output that is fed to the algorithm, and clearly a way to reach a perfect matching between simulation and experiment (and that's because the optimization was done on the difference between both values, could arbitrarily high accuracies be achieved otherwise?). I might be missing a point here, but to remove my concerns and demonstrate an objective approach, I would suggest a comparison of the experimental results without correction, with only phase correction, with only amplitude correction, and with both phase and amplitude correction. I think the phase compensation should lead to the right results, obviously with lower output power due to losses, yet the inference should be correct if the approach works well. That's the test that in my opinion this approach needs to pass.

- How does running an error compensation algorithm impact the claims on speed? This post processing very much impacts the throughput.

- The authors could enhance the discussion by adding possible pathways to reduce the errors.

On Energy and speed:

- Assuming 100 GHz is a long shot. An electro optical conversion is still needed to load the data, what phase modulators would allow such high rates?

- The energy consumption is not properly discussed, saying that it only consumes energy in the data loading and detection doesn't mean much, especially since those are, precisely, very energy consuming processes. The phase shifters will take a good share and of course, a very energy consuming laser is needed, which is usually omitted when discussing optical approaches. Please provide a proper discussion on this aspect. The passive core clearly helps but more information and quantitative analysis would benefit the paper.

Minor point:

- The use of the term slot was confusing. Only until Fig. 2 is that I realized that the "slot" is actually filled with silicon dioxide when somehow the term and the flow of the text seems to suggest it was air. Clarifying this in line 46 would be ideal.'

- The paper Nature communications 10 (1), 1-7, 2019 must be cited! That paper showed the community how to carry out wavefront shaping with on-chip metasurfaces core concept to this approach.

Reviewer #3 (Remarks to the Author):

The paper by Fu et al. contains an interesting study of the use of on-chip diffractive elements to deliver a form of optical neural network. The work is essentially an experimental implementation of the previous simulation/theoretical study published by the same authors in Optics Express last year (and appropriately cited in the current paper as ref. 22).

Although the experimental implementation of the idea first put forward in that previous paper is obviously a very significant achievement, the fact that the ideas in the current paper by Fu et al. are per se not new, means that the novelty of the work is somewhat diminished.

Moreover, the system implemented appears to be a linear optical system, and if so its processing abilities will be limited to those solvable by linear techniques. Indeed, the fact that diffractive optical systems are inherently linear has been a topic of much discussion in the community recently (see e.g. the discussions relating to the paper by Lin et al. - ref 19 in the current work – at www.science.org/doi/10.1126/comment.719040/full/). So, the lack of any non-linear capabilities in the current work is also a factor that limits its novelty and potential applications.

The potential processing speed of the implemented system is stated as being around 2×10^{16} FLOPS, which is indeed very high. However, this assumes a photodetector bandwidth of 100 GHz, which is quite high, and further information on where such a figure was obtained would be useful. More importantly, it is not speed alone that determines the usefulness of a processing system, we should also take into account the processor area. So, a more useful quantitative measure of

processor capabilities is computational density, i.e. speed (in FLOPS or MAC/s) divided by processor area (in mm² or similar). Such a measure should be used in this case, and quantitative comparisons made for example to other linear optical processors as described in the recent Nature journal focus on the topic (see <https://www.nature.com/articles/d41586-020-03572-y> and the associated papers).

Also, in relation to the speed, a limiting factor in a real-world implementation is likely to be the electronic and optical/electronic interfaces needed in the implementation, in particular the DAC shown in Fig 5. Additional information on such limitations, and how they might be mitigated, should be included.

Turning now to energy consumption, for meaningful comparisons of different systems a commonly used quantitative measure is the energy required for one FLOP or MAC (or FLOPS/W). It is recommended that such a measure be used in the current work (e.g. in Table 1). Also Table 1 should include a performance comparison with other photonic technologies, such as the TPUs mentioned previously (in Nature 2021) and indeed common electronic implementations (Google's TPU, NVIDIA chips etc).

Finally, it is noted that the accuracy of the classification tasks carried out is low, prior to application of the compensation algorithm. The errors mainly arise, it seems, from fabrication tolerances/errors affecting the optical phase of light propagating through the system. Further discussion in the main text on how such phase errors might be reduced, rather than compensated, would be beneficial.

Response to Referees' Comments

Dear Referees:

Thank you very much for taking time out of your busy schedule to review our manuscript entitled “**Photonic machine learning with on-chip diffractive optics**” (ID: NCOMMS-22-14333A). We sincerely thank all referees for their highly constructive reviews and valuable feedback to improve the quality of our manuscript. In response to these comments and suggestions, we have carried out additional theoretical calculations and experiments, acquired new data, conducted further analysis, and thus strengthened the paper. We have made thorough revisions accordingly and addressed all the concerns made by the reviewers point by point as detailed next. Moreover, all the revisions/additions are marked in blue font in the revised manuscript.

Point-by-Point Responses

To Reviewer # 1:

Reviewer #1 (*Remarks to the Author*):

“The authors proposed a on-chip diffractive optical neural network (DONN) based on a silicon-on-insulator (SOI) platform to achieve machine-learning computing with light-speed, high integration, and efficiency. The authors fabricated 1-16 hidden-layer and 3-hidden-layer on-chip DONNs and perform a classification task on the Iris dataset with the accuracies of 86.7% and 90%. Although the authors have obtained good classifying performance, I strongly concern the novelty and significance of the manuscript. In fact, the proposed on-chip DONN architecture based on the integrated one-dimensional dielectric metasurface is very similar to that in [Ref. 33]. The novelty declared by the authors are mainly compared to D2NNs based on 3D printing and diffractive propagations in space. However, there have been a lot of researches on planar optical D2NN and on-chip D2NN. The authors do not address these highly related works to illustrate the advantages of this work. Hence I suggest to reject its publication in Nature Communications.”

Extract 1 from the above remarks: “In fact, the proposed on-chip DONN architecture based on the integrated one-dimensional dielectric metasurface is very similar to that in [Ref. 33].”

Response to Extract 1: We sincerely thank you for the comment. Although there are similarities between the two works, there are also differences, and the differences are still of great value for scientific research. We hope that the different implementation methods of the whole neural network system can bring different attention and inspiration to readers. Here, we would like to further explain the differences in the following aspects.

First of all, the on-chip electromagnetic propagation model in our work is different from that of [Ref. 33]. In our work, the on-chip electromagnetic propagation (OEP) model is modified based on the Huygens-Fresnel principle under restricted propagation conditions, which can be described by Eq. (R1):

$$w_{p,q}^m = \frac{1}{j\lambda} \cdot \left(\frac{1 + \cos \theta_{p,q}}{2r_{p,q}} \right) \cdot \exp\left(j \frac{2\pi r_{p,q} n_{slab}}{\lambda} \right) \cdot \eta \exp(j\Delta\phi) \quad (\text{R1})$$

where m represents the m -th layer of the network, p represents the p -th neuron located at (x_p, y_p) of layer m , q represents the q -th neuron located at (x_q, y_q) of layer $m - 1$,

λ is the working wavelength, $j = \sqrt{-1}$ is an imaginary unit, $\cos \theta_{p,q} = (x_p - x_q)/r_{p,q}$,

$r_{p,q} = \sqrt{(x_p - x_q)^2 + (y_p - y_q)^2}$ is the distance between the q -th neuron in layer $m - 1$ and the p -th neuron in layer m , n_{slab} is the effective refractive index (ERI) of the slab waveguide, η is a specific coefficient of the amplitude and $\Delta\phi$ is a fixed phase delay.

The η and $\Delta\phi$ are the amplitude and phase correction factors of light propagation in the planar waveguide, which are obtained by referring to the 2.5D variational finite-difference time-domain (FDTD) simulation results. Fig. R1 is a comparison of the calculation results of Eq. (R1) and 2.5D variational FDTD after the light propagates in a 300 μm wide waveguide for a distance of 250 μm . It is not difficult to see that the calculation results of the two models are highly consistent. However, as for the OEP

model, there is no discussion and comparison of the calculation results between its model and FDTD simulation in [Ref. 33].

Figure R1. **a, b** The field intensity and phase distribution of the input signal, respectively. **c, d** The field intensity and phase distribution of the input signal propagating 250 μm later in a slab waveguide, the results are calculated by the modified OEP model

Eq. (1) (black line) and 2.5D variational FDTD (red line), respectively.

Second, the value mapping model of the neurons in our work is different from that of [Ref. 33]. Although our work (three slots filled with silicon dioxide as a group) and the [Ref. 33] (two slots filled with silicon dioxide as a group) both adopt the slot-group structure to complete the value mapping of the neurons. However, the specific implementation methods are different. We fully considered the preconditions that need to be met under the approximation conditions and discussed them in detail, as shown in Fig. R2. For example, the effective refractive index (ERI) of the slot area used in the training process is 2.166, thus when the physical structure needs to be realized, the ERI should be a constant (2.166) to meet the requirement that the parameters obtained in the advanced training process are consistent with those obtained after the actual chip structure is fabricated. Here, in the Fig. R2, it is apparent to see that when the slot-group exceed two number of slots and the input angle between the input light source and the

slot-group is small, the ERI of the slot area (slot-group of the physical structure) is calculated around 2.166 (calculated by FDTD), which can ensure the correctness of the mapping process. Otherwise, the ERI of the slot area is not around the constant 2.166, then it cannot meet the accurate value mapping of the pretrained neurons. For more details, please refer to the Supplementary note 1.3 or our previous published works [1].

Figure R2. **a** Calculation of the ERI under a different number of identical slots in a silicon slot group, here each of the slot is filled with silicon dioxide. The incident light source is flat light, and the center distance of the adjacent slots is 500 nm, the width of the slots is 200 nm, the thickness is 220 nm, and the length is 1.964 μm . The length of 1.964 μm is randomly generated. **b** The incident

light source is a point light, and the distance between the light source and the HL is 30 μm , 250 μm , and 450 μm , respectively. Among them, the HL consists of 100 identical slots, numbered from 1 to 100. The dots are the ERI values calculated by the phase delay $\Delta\varphi_i$ generated by the light passing through the slot of the corresponding sequence number. n_{eff0} is the ERI in the pre-training process, the period of the slots is 500 nm, the width of the slots is 200 nm, the thickness is 220 nm, and the length is 2 μm .

As for the value mapping model of the neurons, we have discussed the whole mapping process in detail under the approximate conditions, including the change of the ERI by the slot-group composed of different slot numbers, and the change of the ERI of the slot-group when the incident light source enters the slot-group with different angles. However, there is no specific discussion on the value mapping model of the neurons in [Ref. 33].

Third, it is imperatively significant for the on-chip diffractive optical neural network (DONN) system to have the ability to error correction. Error correction can not only improve the processing fault tolerance but also improve the robustness of the system, which is an important premise and guarantee to ensure that the DONN system can work effectively. It is worth emphasizing that the compensation for system

errors is one-time, once the error compensation is completed, it would not bring additional consumption to the work of the system. In our work, we propose a detailed scheme of system error compensation and give experimental verification, which is not mentioned in [Ref. 33]. In [Ref. 33], it is proposed to consider the errors brought by the fabrication in the pre-training process to improve the robustness of the chip performance. We believe this is a good method to improve the robustness of the chip fabricating process, and we will consider this method in future works. **However**, despite this, the requirements for the chip fabricating process are very high (the error caused by machining may still exceed the error range assessed in advance during the pre-training stage), and ulteriorly, it is necessary to clearly understand the extent to which the fabricating errors may affect the phase, which is also not easy. **Furthermore**, in addition to the chip fabricating errors, the signal loading and signal detection stage would also bring errors to the whole system, which are more difficult to be assessed in advance during the pre-training process. Consequently, we believe that the error compensation scheme of the whole system is of great significance and indispensable.

Fourth, the input signal loading method is different. In our work, the input signal loading device is on-chip and only includes a phase shifter (PS) and its driver. While, in [Ref. 33], the input signal loading device is off-chip and consists of a beam expander (BE), a programmed micromirror device (DMD) and its driver, a beam splitter (BS), and a Lens group. In terms of portability and application scenarios, on-chip signal loading devices may be better. In terms of the modulation rate of signal loading, the current highest modulation rate of DMD is K Hz, while the on-chip modulator can achieve G Hz, so the on-chip signal loading is more beneficial to the calculation speed and power consumption of the whole system. In terms of system stability, the on-chip signal loading can avoid the errors caused by external optical path calibration.

Fifth, the output signal detecting area is different. In our work, the output signal is coupled first into an inverse taper with a width of 8 μm (one of the super parameters of the DONN system), then to a single-mode waveguide, which is more tolerant to the

errors brought by the chip fabricating process. While in [Ref. 33], the output signal is coupled directly into a single-mode waveguide, the ability to resist fault tolerance would be reduced.

Last but not least, the specific DONN system structure is different. For example, we have connected an inverse tape to the output interface of the DONN system structure, which can prevent the reflected light of the output interface from interfering with the system performance, because the reflection of the output interface cannot be considered in the DONN parameter training process. While the [Ref. 33] does not take this into account, which is one of the important differences.

To sum up, there are many differences between our work and the [Ref. 33], and these differences have imperatively important innovative value for scientific research and significant inspiration for readers.

Additionally, in particular, to further improve the significance of our work, we designed an on-chip DONN to resolve a more complex classification task, which is the Modified National Institute of Standards and Technology (MNIST) handwritten digit database. **We not only did the simulation test on the MNIST dataset but also experimentally demonstrated the performance of the MNIST classification chip.** The accuracy of simulation and experiment is 96.3% and 86.0%, respectively. The corresponding content has been added to the revised manuscript and marked in blue font.

In fact, the cited article [Ref. 33] in our resubmitted manuscript (**NCOMMS-22-14333A**) is published online on April 19, 2022. While the first draft of our article (**NCOMMS-22-14333**) was submitted on April 12, 2022, which was submitted 7 days earlier than the online publication of the article [Ref. 33]. Therefore, these two works are carried out independently in time and space. The reason why [Ref. 33] appears in the list of the article (**NCOMMS-22-14333A**) is that [Ref. 33] has been published online when we submitted the manuscript for the second time (5 May 2022), thus we

added [Ref. 33] to the list of references. We deem that our work and the [Ref. 33] are completely independent and have their own characteristics.

Extract 2 from the above remarks: *“The novelty declared by the authors are mainly compared to D2NNs based on 3D printing and diffractive propagations in space. However, there have been a lot of researches on planar optical D2NN and on-chip D2NN. The authors do not address these highly related works to illustrate the advantages of this work.”*

Response to Extract 2: Thanks for the reviewer’s valuable comment. Presently, we think that the work of D²NN with on-chip experimental verification is only the [Ref. 33] cited in the submitted manuscript. The differences between our work and the [Ref. 33] have been described in detail in **“Response to Extract 1”**.

In addition, in the original submitted manuscript (**ID: NCOMMS-22-14333A**), we have compared our work with the performances of other integrated ONNs in Table 2 (**Line 288 on Page 10**) in the original submitted manuscript, which includes some other D²NN works, for example, the basic units composed of MZI and diffractive cells ^[2]. Here, through careful consideration, we have made a more comprehensive and detailed comparison between our work and other integrated D²NNs in Table R1, and add these comparisons to the Table 2 in the original submitted manuscript. The corresponding changes have been made in the revised manuscript (**Line 350 on Page 13**) marked in **blue front**.

Table R1 | Comparison between the on-chip DONN and other integrated ONNs

Index Works	Basic units	Integration in theory (NBDs/ mm ²)	Operational power (J/FLOP)	Throughput (TOPS)	Verification method
Ref. [1]	SWU	$\sim 9.5 \times 10^3$	N/A	1.96×10^3	Simulation
Ref. [2]	MZI+ diffractive cells	< 20	1.41×10^{-15}	32	Experiment
Ref. [3]	MMI	< 25	3.07×10^{-14}	30	Experiment
Ref. [4]	SWU	$\sim 3.75 \times 10^4$	N/A	1.54×10^4	Simulation
Our work	SWU	$\sim 2 \times 10^3$	1.1×10^{-17}	1.38×10^4	Experiment

In Table R1, MZI: Mach-Zehnder interferometer, MMI: Multimode interferometer, SWU: Sub-wavelength unit, NBDs: Number of the basic devices, FLOP: floating-point operation, TOPS: Trillions (10^{12}) of operations per second, FLOPS: floating-point operations. For the operational power, it only refers to the energy consumption of the computing units, the laser, and the signal loading stage when the system is working, excluding the energy consumption of signal detection stage and peripheral driving part. For the throughput, which are uniformly calculated according to Eq. (5) in the revised manuscript. Here, the Eq. (5), $R = 2m \times N^2 \times 10^{11}$ FLOPS (floating-point operations per second) in the revised manuscript is given again. In addition, an example is given here for the calculation of the operational power (J/FLOP). For example, in Ref. [1], there is two 70×70 matrices in the calculation process, thus the calculation result of the total operands is $2 \times 2 \times 70 \times 70 \times 10^{11} = 1.96 \times 10^{15}$ FLOPS. In Ref. [2], there is one 16×10 matrices in the calculation process, thus the calculation result of the total operands is $2 \times 1 \times 16 \times 10 \times 10^{11} = 3.2 \times 10^{13}$ FLOPS. In Ref. [3], there are six 5×5 matrices in the calculation process. In Ref. [4], there are two 196×196 matrices in the calculation process.

Line 288 on Page 10 in original submitted manuscript:

Table 2 | Comparison between the on-chip DONN-3 and other integrated ONNs

Study	Node type	Size of HL (N)	Integration in theory (neurons/ mm^2)	Power consumption of computing part
Ref. [5]	MZI	4	< 10	1.92 W
Ref. [6]	MR covered with PCM	4	< 5	Passive
Ref. [7]	MZI	6	< 10	3.92 W
Ref. [2]	MZI	10	< 20	7.7 mW
Ref. [3]	MMI	5	< 25	2.04W
Our work	Subwavelength unit	186	~ 2000	Passive

Line 350 on Page 13 in revised manuscript:

Table 2 | Comparison between the on-chip DONN-3 and other integrated ONNs

Index Works	Basic units	Integration in theory (NBDs/ mm^2)	Operational power (J/FLOP)	Throughput (TOPS)	Verification method
Ref. [5]	MZI	< 10	7.66×10^{-14}	6.4	Experiment
Ref. [7]	MZI	< 10	2.14×10^{-13}	21.6	Experiment
Ref. [2]	MZI+ diffractive	< 20	1.41×10^{-15}	32	Experiment
Ref. [3]	MMI	< 25	3.07×10^{-14}	30	Experiment
Ref. [8]	MRR covered with PCM	< 5	5.9×10^{-15}	28.8	Experiment
Ref. [1]	SWU	$\sim 9.5 \times 10^3$	N/A	1.96×10^3	Simulation
Ref. [4]	SWU	$\sim 3.75 \times 10^4$	N/A	1.54×10^4	Simulation
Our work	SWU	$\sim 2 \times 10^3$	1.1×10^{-17}	1.38×10^4	Experiment

In Table 2, MZI: Mach-Zehnder interferometer, MRR: Micro-ring resonators, MMI: Multimode interferometer, SWU: Sub-wavelength unit, NBDs: Number of the basic devices, FLOP: floating-point operation, TOPS: Trillions (10^{12}) of operations per second. For the operational power, it only refers to the energy consumption of the computing units, the laser, and the signal loading stage when the system is working, excluding the energy consumption of signal detection stage and peripheral driving part. For the throughput, which are uniformly calculated according to Eq. (5) in the revised manuscript.

Comment 1. The simulation results presented in Figure 4(b)(d)(f) show excessive peaks at the output plane, especially in the proximity of the detectors, which may hinder the system performance considering fabrication margin. I would like to know would the unwanted peaks interference with the detector outputs and could they be eliminated in the training process of the proposed DONN?

Response to comment 1: Thanks for the comment. First of all, the light field (unwanted peaks) distributed near the detection area will not interfere with the detector outputs, because all the light in the output interface is collected through different inverse types before the next step of processing. For example, the light in the detection area is collected by a specific inverse type and enters the single-mode waveguide, and then is coupled into the single-mode fiber by the grating coupler, while the unwanted peaks near the detection area are collected by another specific inverse types and enter the optical attenuation structure and disappear, so the peaks near these detectors will not affect the performance of the system. Second, in the current work, the number of modulating units has been determined, which is limited, thus the unwanted peaks cannot be eliminated in the training process of the proposed DONN. **Theoretically**, the metasurface can realize arbitrary regulation of the light field ^[9]. However, this has high requirements for the number of modulating units. For example, the task of DONNs proposed by our work is to realize a three-classification task, thus we only select three detection areas with 8 μm width for training, while other areas are not considered in the training process. The reason why other regions are not considered is that on the one hand, the distribution of unwanted peaks in these regions will not affect the classification performance of the DONN, thus it is unnecessary to consider other regions. On the other hand, if the distribution of light field in other regions is considered, a better classification result cannot be achieved under the limited existing number of modulating units. Therefore, to obtain better performances of a DONN, a suitable output region according to the actual classification tasks is better to be determined in advance. To sum up, in the current work, the number of modulating units has been

determined, thus the unwanted peaks cannot be eliminated in the training process of the proposed DONN, but it would not affect the performance of the proposed DONNs.

Comment 2. The Iris plants dataset is a widely used benchmark for simple machine learning algorithms, which is easily to achieve high classification accuracy because the feature of input values could be clustered at low feature plane. MNIST handwritten digit database is suggested to be numerically demonstrated.

Response to comment 2: Thanks for the reviewer’s valuable comment and suggestion. We strongly agree with the suggestion, therefore, we not only numerically calculated and verified the classification task of the MNIST handwritten digit database through our designed DONN system but also experimentally demonstrated the on-chip DONN’s performance.

Theoretical calculation part. This part is added in Supplementary note 2.2 in the revised manuscript.

Table R2 indicates the prediction accuracy of the on-chip DONNs for classification task of the MNIST handwritten digit images. The letter “n” means the number of HLs in the on-chip DONN (e.g., the DONN-M3 means the DONN for the MNIST handwritten digit classifier includes three HLs). Figs. R3a is the convergence process of the loss function in the training process of on-chip DONN-M3 and Figs. R3b is the confusion matrixes of blind testing results of on-chip DONN-M3. In the DONN-M3 structure, there are 70 neurons on each HL, and phase values of the three HLs are shown in the form of pixels in Fig. R3c (Layer1, Layer2 and Layer3). The linear arrangement phase values of each HL are recombined and given in the form of a 2×35 matrix. The accuracy of numerical calculation and experiment is 96.3% and 86.0%, respectively.

Supplementary Table R2 | The prediction accuracy (numerical calculation) of various DONNs on the MNIST blind test sets (10000)

DONN with n hidden layers (DONN-Mn)	Accuracy
On-chip DONN-M1	72.6%
On-chip DONN-M2	95.5%
On-chip DONN-M3	96.3%
On-chip DONN-M4	96.5%
On-chip DONN-M5	96.1%

Supplementary Figure R3. a The loss curves on the training set (black line) and the blind testing accuracy curves on the test set (red line) for the optimized DONN-M3 during the learning procedure. **b** The confusion matrix of the MNIST test set obtained by simulation shows that the accuracy of blinding test set is 96.3%. **c** The phase profile of DONN-M3 hidden layer after training, the linear phase profile of each hidden layers is converted to a 2×35 pixelated image, and each rectangular pixel in the figure represents a pre-trained phase value of a neuron.

Experimental verification part. This part is added in the revised manuscript (Line 350 on Page 13) and the Supplementary note 4.3 in the revised manuscript.

Further experimental verification. Based on the same design principle of the Iris flower classifier. A more complicated dataset, the Modified National Institute of Standards and Technology (MNIST) handwritten digit images, is used to validate the functionality of

our proposed on-chip DONN. The MNIST dataset is split into training (60,000 images) and testing sets (10,000 images). In this work, for the handwritten digit classifier, the input 28×28 grayscale image is reshaped into a 784×1 vector and compressed into 10 features through a full connection layer network.

For the optimized on-chip DONN-M3, the length of the HLs was $105 \mu\text{m}$ along the Y-axis, and each HL contained 70 neurons (consisting of 210 rectangular slots). The distance between two successive HLs was also $250 \mu\text{m}$ along the X-axis. The ten input features are loaded onto the ten corresponding input single-mode waveguides and propagate directly into the slab waveguide, and then propagate $250 \mu\text{m}$ through the slab waveguide to reach the first HL. After light exits the last HL, it also propagates $250 \mu\text{m}$ until it reaches the output layer of the network, with ten detector regions “ D_i ” ($i = 1, 2, \dots, 10$) arranged in a linear configuration. Each detector region is assigned a specific

category. The width of each detector region was $8\ \mu\text{m}$, and the distance between the centers of the two neighboring detector regions was $8\ \mu\text{m}$.

Fig. R4. **a, b,** and **c** The pictures after wiring and packaging. **d** The micrograph of the on-chip DONN-M3, in which the yellow dotted line frame part is the chip structure of this experiment. **e** The SEM picture of the diffractive units. **f** Confusion matrix of the experimental results of DONN-M3. **g, h,** and **i** Recognition result of handwritten digits “3”, “5” and “9”, respectively

We randomly selected 100 handwritten digits from the 10000 blind test sets for experimental verification, achieving a classification accuracy of 86.0% under the external error compensation. The relevant pictures during the packaging process of the on-chip DONN-M3 are shown in Fig. R4 a, b, and c. The micrograph of the DONN-M3 structure is shown in Fig. R4 d, and the close-up taken by scanning electron microscope (SEM) of the diffractive units is shown in Fig. R4 e. The confusion matrix of the experimental testing result is shown in Fig. R4 f. The recognition results of handwritten digits “3”, “5” and “9” after system error compensation are shown in Fig. R4 g, h, and i.

***Comment 3.** The 1D nature of the proposed platform highly limits the classification of high-resolution images. Can the authors discuss this limitation and compare it with 2D metasystems?*

Response to comment 3: Thanks for the comment. For high-resolution images, the input features (dimensions) are often high. The problem of high-dimensional input can be solved by the lens group for the 2D metasystem. However, for the 1D metasystem, it is difficult to input high-dimensional signals one time due to the limitation of its own physical structures. Therefore, the 1D metasystem needs to do some special processing in advance when dealing with the high-resolution images or other tasks with high-dimensional characteristics.

For the limited input dimension of the 1D metasystem, the problem can be solved from two aspects. First, by designing special physical structures to improve the input dimensions of the 1D metasystem, such as realizing the mapping of input signals in

different modes through the special design of waveguide structure^[10]; The other way is by realizing dimensionality reduction. For example, adopt appropriate algorithms to reduce the dimension of the original signal, and then input the reduced low dimension signal to the 1D metasystem to complete the subsequent tasks^[2]. At present, there are many methods for dimensionality reduction^[11-16]. Of course, if the higher efficiency (higher speed and lower consumption) is strictly demanded, the 1D metasystem optical convolution structure could be designed to achieve dimensionality reduction, which can not only obtain faster dimensionality reduction results but also reduce the energy consumption in the dimensionality reduction process. In fact, our team is also working to solve the problem of limited input dimensions of the 1D metasystem, and we hope to effectively solve this problem in the near future.

Moreover, the comprehensive comparison between the 1D metasystem and the 2D metasystem is listed in Table R2 according to some published works.

Table R2 | Comparison between 1D metasystem and 2D metasystem

Meta-system	Index Works	Integration in theory (neurons/ mm ²)	Stability/ Portability	Fabrication	Signal input dimension
1D	Ref. [17]	$\sim 6.7 \times 10^3$	Integrated	CMOS Compatible	Low (15 inputs)
	Ref. [1]	$\sim 1.7 \times 10^3$	Integrated	N/A	Low (13 inputs)
	Our work	$\sim 2.0 \times 10^3$	Integrated	CMOS Compatible	Low (10 inputs)
2D	Ref. [18]	~ 10	Discrete device	3D Printed	High
	Ref. [19]	$\sim 1.82 \times 10^6$	Integrated	3D Nano-printed	High
	Ref. [20]	$\sim 6.25 \times 10^2$	Discrete device	CMOS Compatible	High

From the Table R2, as the reviewer pointed out, the 1D nature of the proposed platform highly limits the classification of high-resolution images because of its low signal input dimension, while the 2D metasystem is not limited by the input dimension. However, in addition, the 1D metasystem has the following advantages over the 2D metasystem. **First**, the fabrication of the 1D metasystem is compatible with CMOS process, which is dramatically favorable for large-scale and low-cost manufacturing. **Second**, the

whole structures of the 1D metasytem can be integrated on the same chip, thus its stability, portability, and versatility are better than the 2D metasytem. **Third**, the basic structural unit size of the 1D metasytem is generally small (in the order of nanometers), its integration is generally higher than that of the 2D metasytem. Although the Ref. [19] is an integrated form with a high degree of integration, its process requirements are very high, and its versatility and expansibility are limited. **To sum up**, the 1D metasytem might be superior to the 2D metasytem in integration, stability, portability, versatility and massive manufacturing, thus might have more extensive application fields and scenarios than the 2D metasytem.

Comment 4. I believe that the real system based on the proposed DONN could not realize the theoretical speed given in Line 261. Though the photodetection rate and the equivalent processing speed on the platform of DONN exhibit unlimited potentials, I am afraid that the bottleneck is the digital-to-analog conversion (DAC) stage that modulate the signal for input vector injection. Could the author present some discussion relating to this problem?

Response to comment 4: Thanks for the comment. We would like to discuss the problem from the following aspects. **First of all**, with regard to DACs, according to the existing research study, the all-electronic digital-to-analog converter (DAC) with 100 GHz electrical bandwidth has been demonstrated [21]. **Second**, with regard to modulators, according to the published works, the optical modulators based on thin-film lithium niobate platform support modulation data rate up to 320 Gbit per second [22], the electrode on thin-film LN-on-quartz opens up the possibility of achieving sub-volt modulators while having a 3 dB bandwidth >100GHz [23], and the thin film crystal ion sliced (CIS) LiNbO₃ phase modulator that demonstrated an unprecedented measured electro-optic (EO) response up to 500 GHz [24]. **Third**, for the detectors, there are high detected rates can be estimated exceeding 100 GHz in some published works [25,26]. The performance of these reported DACs, modulators, and detectors can meet the calculation conditions assumed in the submitted manuscript. We believe that with the

continuous improvement of DAC, modulator and detector performance, the computing speed based on the DONN system will continue to be realized in real systems. In addition, we also hope that the high computing speed requirements provided by the DONN system can in turn further promote the performance of DACs, modulators and detectors.

In addition, we need to explain that the figure 2.08×10^{16} FLOPS was calculated incorrectly and we corrected it here. The corrected figure is 1.38×10^{16} FLOPS, which is calculated according to Eq. (5) in the original submitted manuscript. The corresponding changes were also corrected in the revised manuscript (**Line 325 on Page 12**) and marked in blue font. The result (1.38×10^{16} FLOPS) mentioned in the revised manuscript (**Line 325 on Page 12**) is the calculation speed of subsequent processing of the input data implemented by DONNs, which does not take into account the time required for input signal loading. Some related works also have similar introductions [1,4,5].

Line 261 on Page 9 in original submitted manuscript:

“Therefore, for the on-chip DONN-3, the computation speed is approximately 2.08×10^{16} FLOPS calculated by Eq. (5)”.

Line 325 on Page 12 in the revised manuscript:

“Therefore, for the on-chip DONN-3, the computation speed is approximately 1.38×10^{16} FLOPS calculated by Eq. (5)”.

Comment 5. Formulas and figures should be revised for a better comprehension. For example, in Equation (1), the transmission parameter w_i^m would better be replaced by w_{ij}^m ; in Figure 1, the matrix and vectors should be marked accordingly for a contrast with the scalar variables; in Figure (6), ‘Truth Label’ and ‘Predicted Label’ should be added at the corresponding axis.

Response to comment 5: Thanks for the reviewer’s valuable comment and suggestion. For Eq. (1), we have modified the footmark of Eq. (1) (**Line 71 on Page 2**) in the original submitted manuscript. Here, we modified the Eq. (1) and given it in red font,

as shown as Eq R-1.

$$w_{p,q}^m = \frac{1}{j\lambda} \cdot \left(\frac{1 + \cos\theta_{p,q}}{2r_{p,q}} \right) \cdot \exp\left(j \frac{2\pi r_{p,q} n_{slab}}{\lambda} \right) \cdot \eta \exp(j\Delta\phi) \quad (\text{R-1})$$

where m represents the m -th layer of the network, p represents the p -th neuron located at (x_p, y_p) of layer m , q represents the q -th neuron located at (x_q, y_q) of layer $m - 1$,

λ is the working wavelength, $j = \sqrt{-1}$ is an imaginary unit, $\cos\theta_{p,q} = (x_p - x_q)/r_{p,q}$,

$r_{p,q} = \sqrt{(x_p - x_q)^2 + (y_p - y_q)^2}$ is the distance between the q -th neuron in layer $m - 1$ and the p -th neuron in layer m , n_{slab} is the effective refractive index (ERI) of the

slab waveguide, η is a specific coefficient of the amplitude and $\Delta\phi$ is a fixed phase delay^[1]. We have made corresponding changes about Eq. (1) in the revised manuscript (Line 78-85 on Page 2) and marked in blue font.

Line 71 on Page 2 in original submitted manuscript:

$$w_i^m = \frac{1}{j\lambda} \cdot \left(\frac{1 + \cos\theta_i}{2r_i} \right) \cdot \exp\left(j \frac{2\pi r_i n_{slab}}{\lambda} \right) \cdot \eta \exp(j\Delta\phi) \quad (1)$$

Line 78-85 on Page 2 in the revised manuscript:

$$w_{p,q}^m = \frac{1}{j\lambda} \cdot \left(\frac{1 + \cos\theta_{p,q}}{2r_{p,q}} \right) \cdot \exp\left(j \frac{2\pi r_{p,q} n_{slab}}{\lambda} \right) \cdot \eta \exp(j\Delta\phi) \quad (1)$$

where m represents the m -th layer of the network, p represents the p -th neuron located at (x_p, y_p) of layer m , q represents the q -th neuron located at (x_q, y_q) of layer $m - 1$,

λ is the working wavelength, $j = \sqrt{-1}$ is an imaginary unit, $\cos\theta_{p,q} = (x_p - x_q)/r_{p,q}$,

$r_{p,q} = \sqrt{(x_p - x_q)^2 + (y_p - y_q)^2}$ is the distance between the q -th neuron in layer $m - 1$ and the p -th neuron in layer m , n_{slab} is the effective refractive index (ERI) of the slab waveguide, η is a specific coefficient of the amplitude and $\Delta\phi$ is a fixed phase delay.

For Figure 1 (Fig. 1), the notes of Fig.1 (Line 101-106 on Page 3) in the original submitted manuscript have been modified and supplemented (shown as Fig. R5), the matrixes are marked in bold, the other parameters are all scalar variables. This time, we believed that the readers can more clearly understand the meaning of Fig.1. The modification is also marked in red font in the Fig. R5, and the corresponding changes have also been corrected in the revised manuscript and marked in blue front.

Fig. R5. (a) Schematic of an on-chip diffractive optical neural network (DONN), each diffractive unit on a given layer acts as a secondary source of a wave, the amplitude and phase of which are determined by the product of the input wave and the complex-valued transmission at that unit. Each diffractive unit is a silicon slot group composed of three identical silicon slots filled with silicon dioxide and represents a single neuron in the on-chip DONNs. (b) Logic diagram of (a), which mathematically describes the physical calculation process of the on-chip DONN. The formula between Fig. 1 (a) and Fig. 1 (b) is the mathematical expression of DONN, where "T" represents matrix transpose; $\text{diag}(e^{j\phi_{11}}, \dots, e^{j\phi_{1n}})$, $\text{diag}(e^{j\phi_{21}}, \dots, e^{j\phi_{2n}})$, and $\text{diag}(e^{j\phi_{31}}, \dots, e^{j\phi_{3n}})$ refer to a diagonal matrix, that is, a matrix whose elements outside the main diagonal are all 0, where the phase values $(\phi_{11}, \dots, \phi_{1n}, \phi_{21}, \dots, \phi_{2n}, \phi_{31}, \dots, \phi_{3n})$ are generated by the corresponding diffractive units. $W^{(k)}$ represents the k -th diffraction matrix derived from the on-chip electromagnetic propagation model (Eq. (1)). (x_1, x_2, x_3, x_4) is the input; (y_1, y_2, y_3) is the output.

Line 101-106 on Page 3 in original submitted manuscript:

Fig. 1. (a) Schematic of an on-chip diffractive optical neural network (DONN), each diffractive unit on a given layer acts as a

secondary source of a wave, the amplitude and phase of which are determined by the product of the input wave and the complex-valued transmission at that unit. Each diffractive unit is a silicon slot group composed of three identical silicon slots and represents a single neuron in the on-chip DONNs. (b) Logic diagram of (a), which mathematically describes the physical calculation process of the on-chip DONN.

Line 108-118 on Page 3 in the revised manuscript:

Fig. 1. (a) Schematic of an on-chip diffractive optical neural network (DONN), each diffractive unit on a given layer acts as a

secondary source of a wave, the amplitude and phase of which are determined by the product of the input wave and the complex-valued transmission at that unit. Each diffractive unit is a silicon slot group composed of three identical silicon slots filled with silicon dioxide and represents a single neuron in the on-chip DONNs. (b) Logic diagram of (a), which mathematically describes the physical calculation process of the on-chip DONN. The formula between Fig.1 (a) and Fig.1 (b) is the mathematical expression of DONN, where "T" represents matrix transpose; $\text{diag}(e^{j\phi_{11}}, \dots, e^{j\phi_{1n}})$, $\text{diag}(e^{j\phi_{21}}, \dots, e^{j\phi_{2n}})$, and $\text{diag}(e^{j\phi_{31}}, \dots, e^{j\phi_{3n}})$ refer to a diagonal matrix, that is, a matrix whose elements outside the main diagonal are all 0, where the phase values $(\phi_{11}, \dots, \phi_{1n}, \phi_{21}, \dots, \phi_{2n}, \phi_{31}, \dots, \phi_{3n})$ are generated by the corresponding diffractive units. $W^{(k)}$ represents the k -th diffraction matrix derived from the on-chip electromagnetic propagation model (Eq. (1)). (x_1, x_2, x_3, x_4) is the input; (y_1, y_2, y_3) is the output.

For Figure (6) (Fig. 6): ‘Truth Label’ and ‘Predicted Label’ have been added at the corresponding axis, as shown in Fig. R6. The corresponding changes have also been corrected in the revised manuscript (Line 223-227 on Page 9).

Fig. R6. Experimental testing results of on-chip DONN-1 and DONN-3. (a) Confusion matrix of the experimental results of DONN-

1 without compensation. (b) Confusion matrix of the experimental results of DONN-1 after compensation. (c, d) Confusion matrixes of the experimental results of DONN-3 before (c) and after (d) the compensation.

Fig. 6. Experimental testing results of on-chip DONN-1 and DONN-3. (a) Confusion matrix of the experimental results of DONN-1 without compensation. (b) Confusion matrix of the experimental results of DONN-1 after compensation. (c, d) Confusion matrixes of the experimental results of DONN-3 before (c) and after (d) the compensation.

Line 223-227 on Page 9 in the revised manuscript:

Fig. 6. Experimental testing results of on-chip DONN-1 and DONN-3. (a) Confusion matrix of the experimental results of DONN-

1 without compensation. (b) Confusion matrix of the experimental results of DONN-1 after compensation. (c, d) Confusion matrixes of the experimental results of DONN-3 before (c) and after (d) the compensation.

Reviewer #2:

Reviewer #2 (*Remarks to the Author*):

I am in principle supportive of the neat concept the authors have presented in this paper. However, I have serious concerns on the objectivity of the results as a consequence of the error compensation algorithm. Before recommending the publication of this paper, such comments need to be addressed, emphasizing again on demonstrating that the error compensation algorithm is not arbitrarily manipulating the result of the computation. Please see my detailed comments below.

On error compensation

- I am concerned about the way the error compensation algorithm works. For once, the fact that the algorithm allows the authors to reach, exactly, the same values calculated in simulations is somehow a red flag. I can see how the phase errors introduced by fabrication defects can be compensated, but my main problem is the power compensation on the output signals. Since the inference is performed using intensity measurements, a direct manipulation of such intensity is a manipulation of the computing output. Multiplying the three output channels by optimized weights seems like a way to get any expected output that is fed to the algorithm, and clearly a way to reach a perfect matching between simulation and experiment (and that's because the optimization was done on the difference between both values, could arbitrarily high accuracies be achieved otherwise?). I might be missing a point here, but to remove my concerns and demonstrate an objective approach, I would suggest a comparison of the experimental results without correction, with only phase correction, with only amplitude correction, and with both phase and amplitude correction. I think the phase compensation should lead to the right results, obviously with lower output power due to losses, yet the inference should be correct if the approach works well. That's the test that in my opinion this approach needs to pass.

Extract 1 from the remarks: *“I can see how the phase errors introduced by fabrication defects can be compensated, but my main problem is the power compensation on the output signals. Since the inference is performed using intensity measurements, a direct manipulation of such intensity is a manipulation of the computing output. Multiplying the three output channels by optimized weights seems like a way to get any expected output that is fed to the algorithm, and clearly a way to reach a perfect matching between simulation and experiment (and that’s because the optimization was done on the difference between both values, could arbitrarily high accuracies be achieved otherwise?).”*

Response to Extract 1: Thanks for the comment. The purpose of both phase compensation and power compensation is to correct the system error as much as possible. **Once the compensation process is completed, the phase compensation values and power compensation coefficients are uniquely determined (fixed constants), and these compensation constants will not change during the full operation of the whole system.** Of course, in our work, whether phase error compensation or power error compensation, its ability to compensate system error is limited. Only when the two compensation parts cooperate with each other can better compensation effect be achieved. Here, we would like to discuss the system error compensation in detail, and hope to express the problem more clearly and comprehensively.

Firstly, the error of the system mainly comes from three aspects: signal loading, chip fabricating and signal detection stage. The errors caused by the three aspects can be compensated by phase compensation and power compensation. In essence, the phase compensation will change the light intensity distribution in different output areas, and the power compensation is also aimed at the change of the light intensity detected in the output areas. Therefore, the results of the two kinds of compensation are the correction of the output light intensity distribution to correct the system error and make the system perform better.

Next, there is a specific example for the compensation analysis of on-chip DONN errors. Fig. R7a is a 1-hidden-layer DONN, Fig. R7b is the logic diagram of Fig. R7a, and Fig. R7c is the equivalent logic diagram after introducing the system error compensation method. The mathematical calculation process of Fig. R7a is expressed by Eq. (R-2). When the inevitable errors brought by the signal loading and chip fabrication stage, the mathematical calculation process is expressed by Eq. (R-3). To reduce the impacts of system errors on the performance of on-chip DONNs, two steps are adopted to reduce the negative influences. First, the phase compensation is introduced in the signal loading stage, meanwhile, the compensated phases of $(\varphi_{11}, \varphi_{22}, \varphi_{33}, \varphi_{44})$ are generated by the voltages of $(\Delta V_1, \Delta V_2, \Delta V_3, \Delta V_4)$ according to Fig. R8 (Supplementary Fig. S8a). Then, in the signal detection stage, the output power of different ports is correspondingly multiplied by a power compensation factor. The logic diagram of phase compensation and power compensation is shown in Fig. R7c for the specific process. Among them, the phase compensation process is depicted by Eq. (R-4). The power detection process is depicted by Eq. (R-5), and the power compensation stage is expressed by Eq. (R-6).

Fig. R7 a is three-layer DONN with one hidden layer. b is a logic diagram of Fig. R5a. c is the equivalent logic diagram of Fig. R5b after introducing the experimental compensation algorithm.

$$\begin{bmatrix} y_1 \\ y_2 \\ y_3 \end{bmatrix} = \begin{bmatrix} W^{(2)} \end{bmatrix}_{3 \times m} \begin{bmatrix} e^{j\phi_{11}} & 0 & 0 & \dots & 0 \\ 0 & e^{j\phi_{22}} & 0 & \dots & 0 \\ \vdots & \ddots & \ddots & \ddots & \vdots \\ 0 & \dots & 0 & \ddots & 0 \\ 0 & \dots & 0 & 0 & e^{j\phi_{mm}} \end{bmatrix} \begin{bmatrix} W^{(1)} \end{bmatrix}_{m \times 4} \begin{bmatrix} x_1 \\ x_2 \\ x_3 \\ x_4 \end{bmatrix} \quad (\text{R-2})$$

$$\begin{bmatrix} y_1 \\ y_2 \\ y_3 \end{bmatrix} = \begin{bmatrix} W^{(2)} \end{bmatrix}_{3 \times m} \begin{bmatrix} e^{j(\phi_{11} + \phi_{11}^{err})} & 0 & 0 & \dots & 0 \\ 0 & e^{j(\phi_{22} + \phi_{22}^{err})} & 0 & \dots & 0 \\ \vdots & \ddots & \ddots & \ddots & \vdots \\ 0 & \dots & 0 & \ddots & 0 \\ 0 & \dots & 0 & 0 & e^{j(\phi_{mm} + \phi_{mm}^{err})} \end{bmatrix} \begin{bmatrix} W^{(1)} \end{bmatrix}_{m \times 4} \begin{bmatrix} x_1 \\ x_2 \\ x_3 \\ x_4 \end{bmatrix} \quad (\text{R-3})$$

$$\begin{bmatrix} y_1 \\ y_2 \\ y_3 \end{bmatrix} = \begin{bmatrix} W^{(2)} \end{bmatrix}_{3 \times m} \begin{bmatrix} e^{j(\phi_{11} + \phi_{11}^{err})} & 0 & 0 & \dots & 0 \\ 0 & e^{j(\phi_{22} + \phi_{22}^{err})} & 0 & \dots & 0 \\ \vdots & \ddots & \ddots & \ddots & \vdots \\ 0 & \dots & 0 & \ddots & 0 \\ 0 & \dots & 0 & 0 & e^{j(\phi_{mm} + \phi_{mm}^{err})} \end{bmatrix} \begin{bmatrix} W^{(1)} \end{bmatrix}_{m \times 4} \begin{bmatrix} e^{j\phi_{11}} & 0 & 0 & 0 \\ 0 & e^{j\phi_{22}} & 0 & 0 \\ 0 & 0 & e^{j\phi_{33}} & 0 \\ 0 & 0 & 0 & e^{j\phi_{44}} \end{bmatrix} \begin{bmatrix} x_1 \\ x_2 \\ x_3 \\ x_4 \end{bmatrix} \quad (\text{R-4})$$

$$\begin{bmatrix} P_1 \\ P_2 \\ P_3 \end{bmatrix} = \begin{bmatrix} |y_1|^2 \\ |y_2|^2 \\ |y_3|^2 \end{bmatrix} \quad (\text{R-5})$$

$$\begin{bmatrix} P_1' \\ P_2' \\ P_3' \end{bmatrix} = \begin{bmatrix} \alpha_1 P_1 \\ \alpha_2 P_2 \\ \alpha_3 P_3 \end{bmatrix} = \begin{bmatrix} \alpha_1 & 0 & 0 \\ 0 & \alpha_2 & 0 \\ 0 & 0 & \alpha_3 \end{bmatrix} \begin{bmatrix} P_1 \\ P_2 \\ P_3 \end{bmatrix} \quad (\text{R-6})$$

Fig. R8 Experimentally obtained phase modulation curve as a function of the applied voltage

It is not difficult to understand that except the phase errors generated in the chip fabricating stage, the system errors caused by the signal loading and signal detection stage also have a certain impact on the performance of the DONN system. Meanwhile, the power compensation can not only be completed off chip but also be realized on chip by designing amplitude modulation devices (such as MZIs) before the three output terminals. For the phase compensation stage, a set of optimized fixed voltage values ($\Delta V_1, \Delta V_2, \Delta V_3, \Delta V_4$) can be found by the online *in-situ* training through the corresponding algorithm. When this set of fixed voltage values is found, phase

compensation can be realized by adding the original signal voltage value to this set of voltage values; For the power compensation stage, based on the phase compensation result, a set of optimized fixed coefficients ($\alpha_1, \alpha_2, \alpha_3$) can be found through the optimization search algorithm. Finally, by multiplying the output power of all samples by the set of fixed power coefficients, the whole error compensation stage can be completed.

It is worth noting that the compensation ability of these fixed compensation coefficients to the DONN system errors is limited, and the performance of the DONN system cannot be arbitrarily improved through the fixed compensation coefficients.

The compensation analysis of DONN errors is also discussed and analyzed in the original supplementary materials (**Supplementary note 5**), and this time we have added more comprehensive analysis to the corresponding contents in the supplementary materials and marked them in blue font.

Extract 2 from the remarks: *“I would suggest a comparison of the experimental results without correction, with only phase correction, with only amplitude correction, and with both phase and amplitude correction.”*

Response to Extract 2: Thanks for the reviewer’s valuable comment and suggestion. According to the reviewer's suggestions, we conducted additional supplementary experiments, including the experimental without compensation, with only phase compensation, with only power compensation, and with both phase and power compensation. The experimental results are shown in Table R3.

Table R3 | Comparison of test results under different conditions

Ways Accuracy	Without compensation	With only phase compensation	With only power compensation	With both phase and power compensation
DONN-1	56.7%	70.0%	76.7%	86.7%
DONN-3	63.3%	73.3%	73.3%	90%

It is worth noting that only power compensation for system error cannot achieve the

optimal performance, and cannot make the simulation results match the experimental results perfectly, the correction ability is limited. In this work, whether phase error compensation or power error compensation, its ability to compensate system error is limited. Only when the two compensation parts cooperate with each other can better compensation effect be achieved.

Extract 3 from the remarks: *“I think the phase compensation should lead to the right results, obviously with lower output power due to losses, yet the inference should be correct if the approach works well. That’s the test that in my opinion this approach needs to pass.”*

Response to Extract 3: Thanks for the reviewer’s valuable comment. We sincerely agree with your opinion. We believe that phase compensation can effectively correct the system errors. However, we also think that the performance of phase error correction is related to the degree of error caused by chip fabrication. For complex systems, its errors are often more complicated, thus the error correction ability of only the phase compensation this time is limited. In our work, the errors of the whole system come from three aspects, including signal loading, chip fabricating and signal detection stage, and whether phase error compensation or power error compensation, its ability to compensate system error is limited. Only when the two compensation parts cooperate with each other can better compensation effect be achieved. See **Response to Extract 2** for the specific error compensation effect.

Comment 1. *How does running an error compensation algorithm impacts the claims on speed? This post processing very much impacts the throughput.*

Response to comment 1: Thanks for the comment. The whole compensation process is one-time. Once the system error compensation is completed, for example, the compensation parameters described in “**Response to Extract 1**” are obtained through the corresponding compensation methods, the obtained parameters are fixed in the whole system during the working process, and the performance of the DONNs would

no longer be affected. Its throughput and claimed calculation speed would not be impacted.

Comment 2. The authors could enhance the discussion by adding possible pathways to reduce the errors.

Response to comment 2: Thanks for the valuable comment and suggestion. In future work, several methods can be used to reduce the errors. **First**, through using more advanced machining equipment, such as electron beam lithography (EBL), which can fundamentally reduce the errors caused by the chip processing stage. **Second**, the random phase offset with uniform distribution within the interval, for example $(0,0.5\pi)$, can be introduced to each part during the training stage, to improve the system's robustness against nanofabrication variations and free-space phase fluctuations in measurement ^[17]. **Last but not least**, further improve the resolution of the testing instrument and the stability of the testing environment to ensure that the error brought by the testing process is minimized. The sources of system errors and the methods to reduce them have been compensated in the discussion part (Line 289-298 on Page 11) of the original submitted manuscript and marked in blue font.

Line 325 on Page 12 in the revised manuscript:

Sources and solutions of system errors. The error in the system mainly comes from three aspects, including signal loading, chip fabricating, and signal detection stages. In future work, several methods can be used to reduce system errors. First, through using more advanced machining equipment, which can fundamentally reduce the error caused by chip processing. Second, the random phase offset with uniform distribution within the interval, for example $[0,0.5\pi)$, can be introduced to each part during the training stage, such as the signal loading and fabrication part, to improve the system's robustness against nanofabrication variations and free-space phase fluctuations in measurement [33]. Last but not least, it is extraordinarily significant to further improve the resolution of the testing instrument and the stability of the testing environment to ensure that the

error brought by the testing process is minimized.

On Energy and speed:

Comment 3. -Assuming 100 GHz is a long shot. An electro optical conversion is still needed to load the data, what phase modulators would allow such high rates?

Response to comment 3: Thanks for the comment. **First of all**, with regard to phase modulators, according to the published works, the optical modulators based on thin-film lithium niobate platform support modulation data rate up to 320 Gbit per second [22], the electrode on thin-film LN-on-quartz opens up the possibility of achieving sub-volt modulators while having a 3 dB bandwidth $>100\text{GHz}$ [23], the thin film crystal ion sliced (CIS) LiNbO_3 phase modulator that demonstrated an unprecedented measured electro-optic (EO) response up to 500 GHz [24]. **Second**, with regard to DACs, according to the existing research study, the all-electronic digital-to-analog converter (DAC) with 100 GHz electrical bandwidth has been demonstrated [21]. **Third**, for the detectors, there are high detected rates can be estimated exceeding 100 GHz in some published works [25,26]. The performance of these reported DACs, modulators, and detectors can meet the calculation conditions assumed in the submitted manuscript. We believe that with the continuous improvement of DAC, modulator and detector performance, the higher computing speed based on the DONN system will continue to be realized in real systems. In addition, we also hope that the high computing speed requirements provided by the DONN system can in turn further promote the performance of DACs, modulators and detectors.

Comment 4. -The energy consumption is not properly discussed, saying that it only consumes energy in the data loading and detection doesn't mean much, especially since those are, precisely, very energy consuming processes. The phase shifters will take a good share and of course, a very energy consuming laser is needed, which is usually omitted when discussing optical approaches. Please provide a proper discussion on this aspect. The passive core clearly helps but more information and quantitative analysis

would benefit the paper.

Response to comment 4: Thanks for the valuable comment and suggestion. In fact, the energy consumption of laser and phase shifter is quantitatively introduced in the supplementary materials originally submitted (**Supplementary note 7.2: Power consumption**). However, according to the reviewers' suggestions, we feel it is imperatively necessary to make a more detailed supplement to the part of energy consumption to strengthen the paper. The contents of this part are also modified in the supplementary materials and marked in blue font.

First of all, the input power of the laser under $1.55 \mu\text{m}$ is 32 mW. The input signal is loaded by the thermo-optic phase shifters, and the average energy required to set each phase shifter a 2π rad is around 30 mW. In addition to the light source and the signal loading stage, the calculation process of the computing part in our proposed on-chip DONN is fully passive, so the system does not need to consume extra energy in processing the inference tasks. For the proposed DONN-3 system, the operations per second is $R = 1.38 \times 10^{16}$ FLOPS (floating-point operations per second). Therefore, in terms of theoretical calculation, the energy consumed to complete one calculation for the proposed DONN-3 system is about $\frac{(32+4 \times 30) \times 10^{-3} \text{ W}}{1.38 \times 10^{16} \text{ FLOP/s}} = 1.1 \times 10^{-17} \text{ J/FLOP}$.

Moreover, to reduce the power consumption of the phase shifters, one could use doped side heaters^[27,28] or liquid-crystal-based phase shifters^[29].

In addition, for the DONN system, the power consumption of the metasystem is the summation of the power required for propagation and the optical power required to support an optical nonlinearity that could be potential implementations of future devices. Here, the transmission loss of light in the working process of DONN system is ignored, and power consumption is mainly the amount of power required to support an optical nonlinearity. Therefore, assume the saturation power for our saturable absorber is $p \text{ MW/cm}^2$ ($p \approx 1$, e.g. graphene), and an area of a neuron $A = 1.5 \mu\text{m} \times 2.0 \mu\text{m}$, then the total power needed to run the system is estimated to be $P = p \times A \times N = 3N(mW)$, meanwhile, N is the neuron number per layer. For the

proposed DONN-3 system ($N = 186$), the input power of the laser 32 mW and the power required to maintain the operation of the phase shifter 4×30 mW, the operations per second is $R = 1.38 \times 10^{16}$ FLOPS, thus the power consumption is about $\frac{(32+4 \times 30+3 \times 186) \times 10^{-3} \text{ W}}{1.38 \times 10^{16} \text{ FLOP/s}} = 5.15 \times 10^{-17} \text{ J/FLOP}$.

Minor point:

Comment 5. -The use of the term slot was confusing. Only until Fig. 2 is that I realized that the “slot” is actually filled with silicon dioxide when somehow the term and the flow of the text seems to suggest it was air. Clarifying this in line 46 would be ideal.’

Response to comment 5: Thanks for the reviewer’s valuable comment and suggestion.

Line 45-47 in Page 1 in original submitted manuscript: “The 1D dielectric metasurface consists of a series of **silicon slots** and represents the hidden layer (HL) in on-chip DONNs.” In order to define the term more clearly, we modify the expression as follows: The 1D dielectric metasurface consists of a series of **silicon slots filled with silicon dioxide** and represents the hidden layer (HL) in on-chip DONNs. We have modified every similar expression in the original submitted manuscript and marked it in blue font.

Line 48-50 in Page 1 in revised manuscript:

“The 1D dielectric metasurface consists of a series of silicon slots filled with silicon dioxide and represents the hidden layer (HL) in on-chip DONNs.”

Comment 6. - The paper Nature communications 10 (1), 1-7, 2019 must be cited! That paper showed the community how to carry out wavefront shaping with on-chip metasurfaces core concept to this approach.

Response to comment 5: Thanks for the comment and suggestion. We cited this article when discussing neuron value mapping process in the supplementary material before. Now, we have added the citation of this article (Wang, Z. et al. On-chip wavefront shaping with dielectric metasurface. Nature Communications 10(2019).) in the references of the revised manuscript and marked it in blue font.

Furtherly, in particular, to further improve the significance of our work, we designed an on-chip DONN to resolve a more complex classification task, which is the Modified National Institute of Standards and Technology (MNIST) handwritten digit database. We not only did the simulation test on the MNIST dataset but also experimentally demonstrated the performance of the MNIST classification chip. We fabricated a 3-hidden-layer on-chip DONN to resolve the MNIST handwritten digit images classification task, and the accuracy of the blind test dataset obtained in numerical calculation and experimental test is 96.3% and 86.0%, respectively.

Theoretical calculation part. This part is added in Supplementary note 2.2 in the revised manuscript. Table R4 indicates the prediction accuracy of the on-chip DONNs for classification task of the MNIST handwritten digit images. The letter “n” means the number of HLs in the on-chip DONN (e.g., the DONN-M3 means the DONN for the MNIST handwritten digit classifier includes three HLs). Figs. R9a is the convergence process of the loss function in the training process of on-chip DONN-M3 and Figs. R9b is the confusion matrixes of blind testing results of on-chip DONN-M3. In the DONN-M3 structure, there are 70 neurons on each HL, and phase values of the three HLs are shown in the form of pixels in Fig. R9c (Layer1, Layer2 and Layer3). The linear arrangement phase values of each HL are recombined and given in the form of a 2×35 matrix. The accuracy of numerical calculation and experiment is 96.3% and 86.0%, respectively.

Supplementary Table R4 | The prediction accuracy (numerical calculation) of various DONNs on the MNIST blind test sets (10000)

DONN with n hidden layers (DONN-Mn)	Accuracy
On-chip DONN-M1	72.6%
On-chip DONN-M2	95.5%
On-chip DONN-M3	96.3%
On-chip DONN-M4	96.5%
On-chip DONN-M5	96.1%

Supplementary Figure R9. **a** The loss curves on the training set (black line) and the blind testing accuracy curves on the test set (red line) for the optimized DONN-M3 during the learning procedure. **b** The confusion matrix of the MNIST test set obtained by simulation shows that the accuracy of blinding test set is 96.3%. **c** The phase profile of DONN-M3 hidden layer after training, the linear phase profile of each hidden layers is converted to a 2×35 pixelated image, and each rectangular pixel in the figure represents a pre-trained phase value of a neuron.

Experimental verification part. This part is added in the revised manuscript (Line 350 on Page 13) and the Supplementary note 4.3 in the revised manuscript. Based on the same design principle of the Iris flower classifier. A more complicated dataset, the Modified National Institute of Standards and Technology (MNIST) handwritten digit images, is used to validate the functionality of our proposed on-chip DONN. The MNIST dataset is split into training (60,000 images) and testing sets (10,000 images). In this work, for the handwritten digit classifier, the input 28×28 grayscale image is reshaped into a 784×1 vector and compressed into 10 features through a full connection layer network.

For the optimized on-chip DONN-M3, the length of the HLs was $105\ \mu\text{m}$ along the Y-axis, and each HL contained 70 neurons (consisting of 210 rectangular slots). The distance between two successive HLs was also $250\ \mu\text{m}$ along the X-axis. The ten input features are loaded onto the ten corresponding input single-mode waveguides and propagate directly into the slab waveguide, and then propagate $250\ \mu\text{m}$ through the slab waveguide to reach the first HL. After light exits the last HL, it also propagates $250\ \mu\text{m}$ until it reaches the output layer of the network, with ten detector regions “ D_i ” ($i = 1, 2, \dots, 10$) arranged in a linear configuration. Each detector region is assigned a specific category. The width of each detector region was $8\ \mu\text{m}$, and the distance between the centers of the two neighboring detector regions was $8\ \mu\text{m}$.

Fig. R10. a, b, and c The pictures after wiring and packaging. d The micrograph of the on-chip DONN-M3, in which the yellow dotted line frame part is the chip structure of this experiment. e The SEM picture of the diffractive units. f Confusion matrix of the

experimental results of DONN-M3. **g, h, and i** Recognition result of handwritten digits “3”, “5” and “9”, respectively

We randomly selected 100 handwritten digits from the 10000 blind test sets for experimental verification, achieving a classification accuracy of 86.0% under the external error compensation. The relevant pictures during the packaging process of the on-chip DONN-M3 are shown in Fig. R10 a, b, and c. The micrograph of the DONN-M3 structure is shown in Fig. R10 d, and the close-up taken by scanning electron microscope (SEM) of the diffractive units is shown in Fig. R10 e. The confusion matrix of the experimental testing result is shown in Fig. R10 f. The recognition results of handwritten digits “3”, “5” and “9” after system error compensation are shown in Fig. R10 g, h, and i.

Reviewer #3:

Reviewer #3 (*Remarks to the Author*):

The paper by Fu et al. contains an interesting study of the use of on-chip diffractive elements to deliver a form of optical neural network. The work is essentially an experimental implementation of the previous simulation/theoretical study published by the same authors in Optics Express last year (and appropriately cited in the current paper as ref. 22).

Although the experimental implementation of the idea first put forward in that previous paper is obviously a very significant achievement, the fact that the ideas in the current paper by Fu et al. are per se not new, means that the novelty of the work is somewhat diminished.

Moreover, the system implemented appears to be a linear optical system, and if so its processing abilities will be limited to those solvable by linear techniques. Indeed, the fact that diffractive optical systems are inherently linear has been a topic of much discussion in the community recently

(see e.g. the discussions relating to the paper by Lin et al. - ref 19 in the current work – at www.science.org/doi/10.1126/comment.719040/full/). So, the lack of any non-linear capabilities in the current work is also a factor that limits its novelty and potential applications.

The potential processing speed of the implemented system is stated as being around 2×10^{16} FLOPS, which is indeed very high. However, this assumes a photodetector bandwidth of 100 GHz, which is quite high, and further information on where such a figure was obtained would be useful. More importantly, it is not speed alone that determines the usefulness of a processing system, we should also take into account the processor area. So, a more useful quantitative measure of processor capabilities is computational density, i.e. speed (in FLOPS or MAC/s) divided by processor area (in mm^2 or similar). Such a measure should be used in this case, and quantitative comparisons made for example to other linear optical processors as described in the

recent Nature journal focus on the topic (see <https://www.nature.com/articles/d41586-020-03572-y> and the associated papers).

Also, in relation to the speed, a limiting factor in a real-world implementation is likely to be the electronic and optical/electronic interfaces needed in the implementation, in particular the DAC shown in Fig 5. Additional information on such limitations, and how they might be mitigated, should be included.

Turning now to energy consumption, for meaningful comparisons of different systems a commonly used quantitative measure is the energy required for one FLOP or MAC (or FLOPS/W). It is recommended that such a measure be used in the current work (e.g. in Table 1). Also **Table 1** should include a performance comparison with other photonic technologies, such as the TPUs mentioned previously (in Nature 2021) and indeed common electronic implementations (Google's TPU, NVIDIA chips etc).

Finally, it is noted that the accuracy of the classification tasks carried out is low, prior to application of the compensation algorithm. The errors mainly arise, it seems, from fabrication tolerances/errors affecting the optical phase of light propagating through the system. Further discussion in the main text on how such phase errors might be reduced, rather than compensated, would be beneficial.

Extract 1 from the remarks: “The paper by Fu et al. contains an interesting study of the use of on-chip diffractive elements to deliver a form of optical neural network. The work is essentially an experimental implementation of the previous simulation/theoretical study published by the same authors in Optics Express last year (and appropriately cited in the current paper as ref. 22). Although the experimental implementation of the idea first put forward in that previous paper is obviously a very significant achievement, the fact that the ideas in the current paper by Fu et al. are per se not new, means that the novelty of the work is somewhat diminished.”

Response to the Extract 1: Thanks for the comment. The previous published work in *Optics Express* last year is the theoretical exploration of on-chip diffractive optical neural network (DONN), which mainly verified the validity of the value mapping model of the pretrained neurons, all conditions in the simulation process are ideal. The previous simulation/theoretical study is far from and completely different from the actual chip experimental verification. We would like to discuss the significance of the present work from following aspects.

Firstly, we considered and resolved many problems that encountered in the real experiments, such as chip fabricating, boundary reflection of physical structures, signal loading, chip packaging and system testing. While in the simulation process, everything is carried out under ideal conditions, thus the problems in the real experiment are far more complicated.

Secondly, we have proposed a complete and effective online training method to compensate the system errors during the experiment, which can not only improve the fault tolerance of the chip fabrication process but also make the DONN system have a good performance. This is of great practical significance to the DONN system. The significance of this part of work is that the theoretical verification cannot be realized.

Last but not least, compared with the previous works, the task solved by this experiment is more complex. In the simulation article, the UCI heart disease dataset is a two-classification task, while in this experiment, the Iris plants dataset is a three-classification task, which is more complicated. **Furtherly**, in particular, to further improve the significance of our work, we designed an on-chip DONN to resolve a more complex classification task, which is the Modified National Institute of Standards and Technology (MNIST) handwritten digit database. We not only did the simulation test on the MNIST dataset but also experimentally demonstrated the performance of the MNIST classification chip. We fabricated a 3-hidden-layer on-chip DONN to resolve the MNIST handwritten digit images classification task, and the accuracy of the blind

test dataset obtained in numerical calculation and experimental test is 96.3% and 86.0%, respectively.

Theoretical calculation part. This part is added in Supplementary note 2.2 in the revised manuscript. Table R5 indicates the prediction accuracy of the on-chip DONNs for classification task of the MNIST handwritten digit images. The letter “n” means the number of HLs in the on-chip DONN (e.g., the DONN-M3 means the DONN for the MNIST handwritten digit classifier includes three HLs). Figs. R11a is the convergence process of the loss function in the training process of on-chip DONN-M3 and Figs. R11b is the confusion matrixes of blind testing results of on-chip DONN-M3. In the DONN-M3 structure, there are 70 neurons on each HL, and phase values of the three HLs are shown in the form of pixels in Fig. R11c (Layer1, Layer2 and Layer3). The linear arrangement phase values of each HL are recombined and given in the form of a 2×35 matrix. The accuracy of numerical calculation and experiment is 96.3% and 86.0%, respectively.

Supplementary Table R5 | The prediction accuracy (numerical calculation) of various DONNs on the MNIST blind test sets (10000)

DONN with n hidden layers (DONN-Mn)	Accuracy
On-chip DONN-M1	72.6%
On-chip DONN-M2	95.5%
On-chip DONN-M3	96.3%
On-chip DONN-M4	96.5%
On-chip DONN-M5	96.1%

Supplementary Figure R11. a The loss curves on the training set (black line) and the blind testing accuracy curves on the test set (red line) for the optimized DONN-M3 during the learning procedure. **b** The confusion matrix of the MNIST test set obtained by simulation shows that the accuracy of blinding test set is 96.3%. **c** The phase profile of DONN-M3 hidden layer after training, the linear phase profile of each hidden layers is converted to a 2×35 pixelated image, and each rectangular pixel in the figure represents a pre-trained phase value of a neuron.

Experimental verification part. This part is added in the revised manuscript (Line 350 on Page 13) and the Supplementary note 4.3 in the revised manuscript. Based on the same design principle of the Iris flower classifier. A more complicated dataset, the Modified National Institute of Standards and Technology (MNIST) handwritten digit images, is used to validate the functionality of our proposed on-chip DONN. The MNIST dataset is split into training (60,000 images) and testing sets (10,000 images). In this work, for the handwritten digit classifier, the input 28×28 grayscale image is reshaped into a 784×1 vector and compressed into 10 features through a full connection layer network.

For the optimized on-chip DONN-M3, the length of the HLs was $105\ \mu\text{m}$ along the Y-axis, and each HL contained 70 neurons (consisting of 210 rectangular slots). The distance between two successive HLs was also $250\ \mu\text{m}$ along the X-axis. The ten input features are loaded onto the ten corresponding input single-mode waveguides and propagate directly into the slab waveguide, and then propagate $250\ \mu\text{m}$ through the slab waveguide to reach the first HL. After light exits the last HL, it also propagates $250\ \mu\text{m}$ until it reaches the output layer of the network, with ten detector regions “ D_i ” ($i = 1, 2, \dots, 10$) arranged in a linear configuration. Each detector region is assigned a specific category. The width of each detector region was $8\ \mu\text{m}$, and the distance between the centers of the two neighboring detector regions was $8\ \mu\text{m}$.

Fig. R12. a, b, and c The pictures after wiring and packaging. d The micrograph of the on-chip DONN-M3, in which the yellow dotted line frame part is the chip structure of this experiment. e The SEM picture of the diffractive units. f Confusion matrix of the

experimental results of DONN-M3. **g, h, and i** Recognition result of handwritten digits “3”, “5” and “9”, respectively

We randomly selected 100 handwritten digits from the 10000 blind test sets for experimental verification, achieving a classification accuracy of 86.0% under the external error compensation. The relevant pictures during the packaging process of the on-chip DONN-M3 are shown in Fig. R12 a, b, and c. The micrograph of the DONN-M3 structure is shown in Fig. R12 d, and the close-up taken by scanning electron microscope (SEM) of the diffractive units is shown in Fig. R12 e. The confusion matrix of the experimental testing result is shown in Fig. R12 f. The recognition results of handwritten digits “3”, “5” and “9” after system error compensation are shown in Fig. R12 g, h, and i.

Extract 2 from the remarks: *“The lack of any non-linear capabilities in the current work is also a factor that limits its novelty and potential applications.”*

Response to the Extract 2: Thanks for the valuable comment. In fact, the signal detection (optical intensity detection) at the output layer is a way to realize nonlinearity^[7], thus the output layer of our DONN system has the non-linear capability. Although our current work can only realize nonlinearity in the output layer, we can find ways to introduce more nonlinear functions in future research works. Here, we would like to discuss how to combine nonlinear functions with on-chip DONN to make it have more powerful ability to handle complex tasks in future. **First, some active materials exhibit exceptionally high nonlinear responses** (such as two-photon absorption-related free carrier absorption or absorption saturation) and are transparent in the range of telecommunication wavelengths, which can be integrated into the diffractive networks as nanoscale activation functions with solution processing ^[30-32]. **Second, by means of photoelectric fusion, the non-linearity is electrically completed, and the electrically processed information is used as the input of the next on-chip DONN** ^[2,5,32]. Therefore, the non-linearity can be electrically realized in the cascade process of multiple on-chip DONNs, thus realizing the non-linearity of the entire DONN system.

Extract 3 from the remarks: “*The potential processing speed of the implemented system is stated as being around 2×10^{16} FLOPS, which is indeed very high. However, this is assumes a photodetector bandwidth of 100 GHz, which is quite high, and further information on where such a figure was obtained would be useful.*”

Response to the Extract 3: Thanks for the comment. For the detectors, there are high detected rates can be estimated exceeding 100 GHz in some published works [25,26]. In addition, an error needs to be corrected in the original submitted manuscript, the figure 2.08×10^{16} FLOPS was calculated incorrectly and we corrected it here. The corrected figure is 1.38×10^{16} FLOPS, which is calculated according to Eq. (5) in the original submitted manuscript (**Line 258-261 in Page 10**). The similar calculation methods are also introduced in some other published works [1,4,5].

For our designed on-chip DONN-3, which has 186 neurons at each HL, implementing 2 layers of 186×186 matrix multiplication, and assuming that it is operating at a typical 100 GHz photodetection rate, the number of floating-point operations per second (FLOPS) to match the optical network is calculated as: $2 \times 2 \times 186 \times 186 \times 10^{11} = 1.38384 \times 10^{16} \approx 1.38 \times 10^{16}$ FLOPS. The relevant modifications of this part have been made in the revised manuscript and marked in blue font.

Line 258-261 in Page 9 in original submitted manuscript:

$$R = 2m \times N^2 \times 10^{11} \text{ FLOPS} \quad (5)$$

where R is the number of operations per second (the time it takes from receiving input signals to computing an inference result, without considering the time of signal loading stage), which is related to the number m of $N \times N$ matrix, number of neurons N on each HL, and detection frequency of the photodetectors. Therefore, for the on-chip DONN-3, the computation speed is approximately 2.08×10^{16} FLOPS.

Line 320-325 in Page 12 in revised manuscript:

$$R = 2m \times N^2 \times 10^{11} \text{ FLOPS} \quad (5)$$

where R is the number of operations per second (the time it takes from receiving input signals to computing an inference result, without considering the time spent of the

signal loading stage), which is related to the number of $N \times N$ matrix, number of neurons on each HL, and detection frequency of the photodetectors. Therefore, for the on-chip DONN-I3, the computation speed is approximately 1.38×10^{16} FLOPS.

Extract 4 from the remarks: “*More importantly, it is not speed alone that determines the usefulness of a processing system, we should also take into account the processor area. So, a more useful quantitative measure of processor capabilities is computational density, i.e. speed (in FLOPS or MAC/s) divided by processor area (in mm² or similar). Such a measure should be used in this case, and quantitative comparisons made for example to other linear optical processors as described in the recent Nature journal focus on the topic (see <https://www.nature.com/articles/d41586-020-03572-y> and the associated papers).*”

Response to the Extract 4: Thanks for the reviewer’s valuable comment and suggestion. According to the suggestions of the reviewer, we refer to relevant literature and compare our work with other related works. We deem that the computational density is more meaningful for integrated processors, thus we compare our work with the performance of existing integrated ONNs. We compared our work with other integrated ONNs in terms of the footprint, the throughput, and the total number of operands processed per second per square millimeter theoretically. The specific calculation results are shown in Table R6. This part has been added to the Supplementary materials (**Supplementary note 8**) of the revised manuscript and marked it in blue font.

Table R6 | Comparison of partial performances of different optical neural networks (ONNs)

Index Works	Basic device	Footprint (mm ²)	Throughput (TOPS)	Computing capacity in theory (FLOPS/mm ²)
Ref. [5]	MZI	0.68	6.4	9.41×10^{12}
Ref. [7]		0.36	21.6	6×10^{13}
Ref. [2]	MZI and diffractive cells	1.2	32	2.67×10^{13}
Ref. [3]	MMI and phase modulators	2.12	30	1.42×10^{13}
Ref. [8]	MRR covered with PCM	6.07	28.8	1.8×10^{12}
Our work DONN-3	SWU	0.3	1.38×10^4	4.6×10^{16}

Extract 5 from the remarks: “Also, in relation to the speed, a limiting factor in a real-world implementation is likely to be the electronic and optical/electronic interfaces needed in the implementation, in particular the DAC shown in Fig 5. Additional information on such limitations, and how they might be mitigated, should be included.”

Response to the Extract 5: Thanks for the comment. We agree with the referee sincerely. In terms of the real system based on the proposed DONN, whether in the input signal modulation stage or the output signal detection stage, the operation speed of the DONN in a real-world implementation would inevitably be limited. Fortunately, the performance of current DACs, modulators and detectors has also been greatly improved. **First of all**, presently, with regard to DACs, according to the existing research study, the all-electronic digital-to-analog converter (DAC) with 100 GHz electrical bandwidth has been demonstrated ^[21]. **Second**, with regard to modulators, according to the published works, the optical modulators based on thin-film lithium niobate platform support modulation data rate up to 320 Gbit per second ^[22], the electrode on thin-film LN-on-quartz opens up the possibility of achieving sub-volt modulators while having a 3 dB bandwidth >100GHz ^[23], the thin film crystal ion sliced (CIS) LiNbO₃ phase modulator that demonstrated an unprecedented measured electro-optic (EO) response up to 500 GHz ^[24]. **Third**, for the detectors, the high detected rates can be estimated exceeding 100 GHz in some published works ^[25,26]. The performance

of these reported DACs, modulators, and detectors can meet the calculation conditions assumed in the submitted manuscript. We believe that with the continuous improvement of DAC, modulator and detector performance, the computing speed based on the DONN system will continue to be realized in real systems. In addition, we also hope that the higher computing speed requirements provided by the DONN system can in turn further promote the performance of DACs, modulators and detectors.

Extract 6 from the remarks: *“Turning now to energy consumption, for meaningful comparisons of different system a commonly used quantitative measure is the energy required for one FLOP or MAC (or FLOPS/W). It is recommended that such a measure be used in the current work (e.g. in Table 1). Also **Table 1** should include a performance comparison with other photonic technologies, such as the TPUs mentioned previously (in Nature 2021) and indeed common electronic implementations (Google’s TPU, NVIDIA chips etc).”*

Response to the Extract 6: Thanks for the reviewer’s valuable comment and suggestion. According to the suggestions of the reviewer, we made serious thinking and thought that it would be better to integrate this part (including the energy required for one FLOP and the performance comparison with other photonic technologies) into Table 2 (**Line 288 in Page 10**) in the submitted manuscript. The modified table is shown in Table R7. The contents of this part have been replaced in the original submitted manuscript and marked in blue font. In addition, to compare with other photonic technologies and indeed common electronic implementations on the performance of the throughput and operational power ($J/FLOP$), we made another comparison in Table R8 and added this part to the supplementary material (**Supplementary note 9**) and marked in blue font.

Table R7 | Comparison between the on-chip DONN and other integrated ONNs

Index Works	Basic units	Integration in theory (NBDs/mm ²)	Operational power (J/FLOP)	Throughput (TOPS)	Verification method
Ref. [5]	MZI	< 10	7.66×10^{-14}	6.4	Experiment
Ref. [7]	MZI	< 10	2.14×10^{-13}	21.6	Experiment
Ref. [2]	MZI+ diffractive	< 20	1.41×10^{-15}	32	Experiment
Ref. [3]	MMI	< 25	3.07×10^{-14}	30	Experiment
Ref. [8]	MRR covered with PCM	< 5	5.9×10^{-15}	28.8	Experiment
Ref. [1]	SWU	$\sim 9.5 \times 10^3$	N/A	1.96×10^3	Simulation
Ref. [4]	SWU	$\sim 3.75 \times 10^4$	N/A	1.54×10^4	Simulation
Our work	SWU	$\sim 2 \times 10^3$	1.1×10^{-17}	1.38×10^4	Experiment

In Table R7, MZI: Mach-Zehnder interferometer, MRR: Micro-ring resonators, PCM: phase-change material, MMI: Multimode interferometer, SWU: Sub-wavelength unit, TOPS: Trillions (10^{12}) of operations per second, FLOP: floating-point operations. For the TOPS, which are uniformly calculated according to Eq. (5) in the revised manuscript. For the operational power, it only refers to the energy consumption of the computing units, the laser, and the signal loading stage when the system is working, excluding the energy consumption of signal detection stage and peripheral driving part.

Table R8 | Comparison of DONN-3 with other research works and commercial products in terms of Throughput and Operational power consumption

Index Works	Throughput (TOPS)	Operational power (J/FLOP)
Ref. [5]	6.4	7.66×10^{-14}
Ref. [7]	21.6	2.14×10^{-13}
Ref. [2]	32	1.41×10^{-15}
Ref. [3]	30	3.07×10^{-14}
Ref. [8]	28.8	5.9×10^{-15}
Google TPU [33]	23	2.15×10^{-13}
NVIDIA Tesla T4 [34]	130	5.4×10^{-13}
HUAWEI Ascend 910 [35]	640	5.45×10^{-13}
Our work	1.38×10^4	1.1×10^{-17}

Line 288 in Page 10 in the original submitted manuscript:

Table 2 | Comparison between the on-chip DONN-3 and other integrated ONNs

Study	Node type	Size of HL (N)	Integration in theory (neurons/ mm^2)	Power consumption of computing part
Ref. [5]	MZI	4	< 10	1.92 W
Ref. [6]	MR covered with PCM	4	< 5	Passive
Ref. [7]	MZI	6	< 10	3.92 W
Ref. [2]	MZI	10	< 20	7.7 mW
Ref. [3]	MMI	5	< 25	2.04W
Our work	Subwavelength unit	186	~ 2000	Passive

where N is the maximum number of neurons in the HL.

Extract 7 from the remarks: *“Finally, it is noted that the accuracy of the classification tasks carried out is low, prior to application of the compensation algorithm. The errors mainly arise, it seems, from fabrication tolerances/errors affecting the optical phase of light propagating through the system. Further discussion in the main text on how such phase errors might be reduced, rather than compensated, would be beneficial.”*

Response to the Extract 7: Thanks for the valuable comment and suggestion. In the future work, several methods can be used to reduce the errors. **First**, through using more advanced machining equipment, such as electron beam lithography (EBL), which can fundamentally reduce the error caused by chip processing. **Second**, the random phase offset with uniform distribution within the interval, for example $[0, 0.5\pi]$, can be introduced to each part during the training stage, to improve the system’s robustness against nanofabrication variations and free-space phase fluctuations in measurement ^[17]. **Last but not least**, the error of the system comes from three aspects: signal loading, chip fabricating and signal detection stage. Therefore, it is extraordinarily significant to further improve the resolution of the testing instrument and the stability of the testing environment to ensure that the error brought by the testing process is minimized. The sources of system errors and the methods to reduce them have been compensated in the discussion part of the original submitted manuscript and marked in blue font (Line 289-298 on Page 11).

Line 289-298 on Page 11 in revised manuscript:

Sources and solutions of system errors. The error in the system mainly comes from three aspects, including signal loading, chip fabricating, and signal detection stages. In future work, several methods can be used to reduce system errors. First, through using more advanced machining equipment, which can fundamentally reduce the error caused by chip processing. Second, the random phase offset with uniform distribution within the interval, for example $(0, 0.5\pi)$, can be introduced to each part during the training stage, such as the signal loading and fabrication part, to improve the system's robustness against nanofabrication variations and free-space phase fluctuations in measurement^[17]. Last but not least, it is extraordinarily significant to further improve the resolution of the testing instrument and the stability of the testing environment to ensure that the error brought by the testing process is minimized.

Supplementary References

1. Fu, T.Z. *et al.* On-chip photonic diffractive optical neural network based on a spatial domain electromagnetic propagation model. *Optics Express* **29**, 31924-31940 (2021).
2. Zhu, H.H. *et al.* Space-efficient optical computing with an integrated chip diffractive neural network. *Nature Communications* **13**, 1044 (2022).
3. Zhao, X. *et al.* On-chip Reconfigurable Optical Neural Networks. (2021).
4. Zarei, S., Marzban, M.R. & Khavasi, A. Integrated photonic neural network based on silicon metalines. *Optics Express* **28**, 36668-36684 (2020).
5. Shen, Y.C. *et al.* Deep learning with coherent nanophotonic circuits. *Nature Photonics* **11**, 441-446 (2017).
6. Feldmann, J., Youngblood, N., Wright, C.D., Bhaskaran, H. & Pernice, W.H.P. All-optical spiking neurosynaptic networks with self-learning capabilities. *Nature* **569**, 208-214 (2019).
7. Zhang, H. *et al.* An optical neural chip for implementing complex-valued neural network. *Nature Communications* **12**, 1-11 (2021).
8. Feldmann, J. *et al.* Parallel convolutional processing using an integrated photonic tensor core (vol 589, pg 52, 2021). *Nature* **591**, E13-E13 (2021).
9. Yu, N.F. *et al.* Light Propagation with Phase Discontinuities: Generalized Laws of Reflection and Refraction. *Science* **334**, 333-337 (2011).
10. Sunada, S. & Uchida, A. Photonic neural field on a silicon chip: large-scale, high-speed neuro-inspired computing and sensing. *Optica* **8**, 1388-1396 (2021).
11. Carreira-Perpinán, M.A. A review of dimension reduction techniques. *Department of Computer Science. University of Sheffield. Tech. Rep. CS-96-09* **9**, 1-69 (1997).

-
12. Cunningham, P. Dimension reduction. in *Machine learning techniques for multimedia* 91-112 (Springer, 2008).
 13. Espadoto, M., Martins, R.M., Kerren, A., Hirata, N.S.T. & Telea, A.C. Toward a Quantitative Survey of Dimension Reduction Techniques. *Ieee Transactions on Visualization and Computer Graphics* **27**, 2153-2173 (2021).
 14. Ray, P., Reddy, S.S. & Banerjee, T. Various dimension reduction techniques for high dimensional data analysis: a review. *Artificial Intelligence Review* **54**, 3473-3515 (2021).
 15. Burges, C.J. Dimension reduction: A guided tour. *Foundations and Trends® in Machine Learning* **2**, 275-365 (2010).
 16. Chao, G.Q., Luo, Y. & Ding, W.P. Recent Advances in Supervised Dimension Reduction: A Survey. *Machine Learning and Knowledge Extraction* **1**, 341-358 (2019).
 17. Wang, Z., Chang, L., Wang, F., Li, T. & Gu, T. Integrated photonic metasystem for image classifications at telecommunication wavelength. *Nature Communications* **13**, 1-8 (2022).
 18. Lin, X. *et al.* All-optical machine learning using diffractive deep neural networks. *Science* **361**, 1004-1008 (2018).
 19. Goi, E. *et al.* Nanoprinted high-neuron-density optical linear perceptrons performing near-infrared inference on a CMOS chip. *Light-Science & Applications* **10**(2021).
 20. Shi, W.X. *et al.* LOEN: Lensless opto-electronic neural network empowered machine vision. *Light-Science & Applications* **11**(2022).
 21. Chen, X. *et al.* All-Electronic 100-GHz Bandwidth Digital-to-Analog Converter Generating PAM Signals up to 190 GBaud. *Journal of Lightwave Technology* **35**, 411-417 (2017).
 22. Xu, M.Y. *et al.* High-performance coherent optical modulators based on thin-film lithium niobate platform. *Nature Communications* **11**(2020).
 23. Kharel, P., Reimer, C., Luke, K., He, L.Y. & Zhang, M. Breaking voltage-bandwidth limits in integrated lithium niobate modulators using micro-structured electrodes (vol 8, pg 357, 2021). *Optica* **8**, 1218-1218 (2021).
 24. Mercante, A.J. *et al.* Thin film lithium niobate electro-optic modulator with terahertz operating bandwidth. *Optics Express* **26**, 14810-14816 (2018).
 25. Vivien, L. *et al.* Zero-bias 40Gbit/s germanium waveguide photodetector on silicon. *Optics Express* **20**, 1096-1101 (2012).
 26. Xia, F.N., Mueller, T., Lin, Y.M., Valdes-Garcia, A. & Avouris, P. Ultrafast graphene photodetector. *Nature Nanotechnology* **4**, 839-843 (2009).
 27. Absil, P.P. *et al.* Silicon photonics integrated circuits: a manufacturing platform for high density, low power optical I/O's. *Optics express* **23**, 9369-9378 (2015).
 28. Masood, A. *et al.* Comparison of heater architectures for thermal control of silicon photonic circuits. in *10th International Conference on Group IV Photonics* 83-84 (IEEE, 2013).
 29. Xing, Y. *et al.* Digitally controlled phase shifter using an SOI slot waveguide with

-
- liquid crystal infiltration. *IEEE Photonics Technology Letters* **27**, 1269-1272 (2015).
30. Wang, F.F. *et al.* Light Emission from Self-Assembled and Laser-Crystallized Chalcogenide Metasurface. *Advanced Optical Materials* **8**(2020).
31. Wang, F.F. *et al.* Controlling Microring Resonator Extinction Ratio via Metal-Halide Perovskite Nonlinearity. *Advanced Optical Materials* **9**(2021).
32. Williamson, I.A.D. *et al.* Reprogrammable Electro-Optic Nonlinear Activation Functions for Optical Neural Networks. *Ieee Journal of Selected Topics in Quantum Electronics* **26**(2020).
33. Jouppi, N.P. *et al.* In-Datacenter Performance Analysis of a Tensor Processing Unit. *44th Annual International Symposium on Computer Architecture (Isca 2017)*, 1-12 (2017).

REVIEWERS' COMMENTS

Reviewer #1 (Remarks to the Author):

I read through the revised manuscript and response letter thoroughly, and found that it was well modified. All concerns in my original comments have been addressed. Based on this, I recommend acceptance of the revised manuscript.

Reviewer #2 (Remarks to the Author):

I am satisfied with the discussion provided by the authors on the nature of the phase and amplitude error correction and their impact in the precision of the result, which was my main criticism. I appreciate their efforts to extensively discuss the impact of their approach and to significantly expand the manuscript accommodating the many suggestions and extra experiments that the reviewers requested. I recommend the publication of the paper in the revised form.

Reviewer #3 (Remarks to the Author):

The authors have carried out very significant additional work and subsequent modifications to the original manuscript that take into account most of my comments/criticisms contained in my original review.

I still have some concerns over the novelty of the work - bearing in mind the overlap between the current paper and the authors own previously published work (ref 22 of original manuscript).

I also have some concern over the lack of intrinsic non-linearity in the optical system (prior to photodetection).

However, the performance of the experimental system - as now detailed in the revised manuscript that includes accepted figures-of-merit such as computational density and energy/flop - is actually very impressive and exceeds other approaches in several respects by orders of magnitude.

So, overall, I feel that the impressive performance capabilities of the authors approach possibly outweighs the remaining drawbacks (as I see them) of the work.